# RG-SAN: Rule-Guided Spatial Awareness Network for End-to-End 3D Referring Expression Segmentation

**Changli Wu**[1,2*], **Qi Chen**[1*], **Jiayi Ji**[1,4*], **Haowei Wang**[3], **Yiwei Ma**[1],
**You Huang**[1], **Gen Luo**[1], **Hao Fei**[4], **Xiaoshuai Sun**[1], **Rongrong Ji**[1†]

[1] Key Laboratory of Multimedia Trusted Perception and Efficient Computing,
Ministry of Education of China, Xiamen University, 361005, P.R. China
[2] Shanghai Innovation Institute, Shanghai, P.R. China
[3] Youtu Lab, Tencent, Shanghai, P.R. China
[4] National University of Singapore

## Abstract

3D Referring Expression Segmentation (3D-RES) aims to segment 3D objects by correlating referring expressions with point clouds. However, traditional approaches frequently encounter issues like over-segmentation or mis-segmentation, due to insufficient emphasis on spatial information of instances. In this paper, we introduce a Rule-Guided Spatial Awareness Network (RG-SAN) by utilizing solely the spatial information of the target instance for supervision. This approach enables the network to accurately depict the spatial relationships among all entities described in the text, thus enhancing the reasoning capabilities. The RG-SAN consists of the Text-driven Localization Module (TLM) and the Rule-guided Weak Supervision (RWS) strategy. The TLM initially locates all mentioned instances and iteratively refines their positional information. The RWS strategy, acknowledging that only target objects have supervised positional information, employs dependency tree rules to precisely guide the core instance's positioning. Extensive testing on the ScanRefer benchmark has shown that RG-SAN not only establishes new performance benchmarks, with an mIoU increase of 5.1 points, but also exhibits significant improvements in robustness when processing descriptions with spatial ambiguity. All codes are available at `https://github.com/sosppxo/RG-SAN`.

## 1 Introduction

3D Referring Expression Segmentation (3D-RES) is an emerging field that segments 3D objects in point cloud scenes based on given referring expressions [24]. Gaining significant attention for its applications in autonomous robotics, human-machine interaction, and self-driving systems, 3D-RES demands a deeper understanding than 3D Referring Expression Comprehension (3D-REC) [5, 73, 1, 75, 70], which focuses only on locating the referring objects via bounding boxes. 3D-RES, on the other hand, requires identifying instances and providing precise 3D masks.

Early 3D-RES approaches [24, 73] adopted a two-stage paradigm, starting with an independent text-agnostic segmentation model for generating instance proposals, followed by linking these proposals with textual descriptions. This paradigm, separating segmentation and matching, proved suboptimal in performance and efficiency. Recent explorations have shifted towards an end-to-end paradigm. For instance, 3D-STMN [65] achieved efficient segmentation by directly matching superpoints with text, while 3DRefTR [43] integrated 3D-RES and 3D-REC into a unified framework using a multi-task

---

*Equal Contribution.
†Corresponding Author.

38th Conference on Neural Information Processing Systems (NeurIPS 2024).

approach, boosting inference in both tasks. Despite these advancements, limitations persist, primarily due to over-reliance on textual reasoning and insufficient modeling of spatial relationships between instances. For example, as shown in Fig. 1, without spatial modeling, it's challenging to understand and correctly segment the intended chair in scenarios involving complex spatial terms like "far away".

To tackle this issue, the core is to assist textual reasoning by modeling the spatial relationships of core instances. By effectively identifying these spatial relationships within expressions, a substantial improvement can be achieved in comprehending spatial arrangements. Nevertheless, this endeavor is not without its challenges. While accurate positional information is crucial for ensuring precise modeling of spatial relationships, accurately regressing instance positions from textual information is far from a simple task. Furthermore, our available positional information is limited to the target instance, leaving us without supervisory signals for other instances referenced in the expression.

To overcome these challenges, we propose the novel Rule-Guided Spatial Awareness Network (RG-SAN), utilizing the spatial information of the target instance for supervision. This enables the network to accurately depict spatial relationships among all text-described entities, thereby significantly enhancing the model's inference and pointing capabilities. RG-SAN consists of two main components: the Text-driven Localization Module (TLM) and the Rule-guided Weak Supervision (RWS) strategy. TLM initially locates all mentioned instances and iteratively refines their positions, ensuring continuous improvement

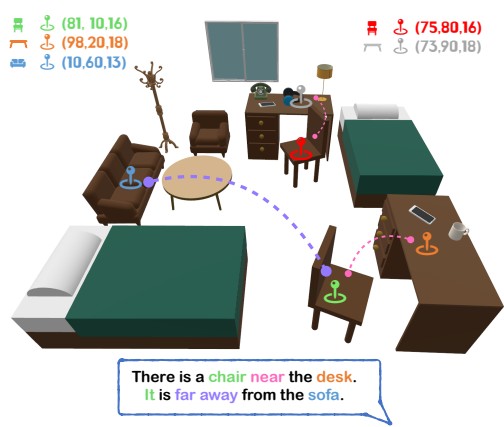

Figure 1: Illustration with a target object and multiple auxiliary objects, associated with a referring expression. The target marked in green represents the main referred instance, while targets in other colors indicate other mentioned entities. This visual highlights the challenge of effectively completing semantic reasoning in the absence of spatial inference.

in location accuracy. RWS, leveraging dependency tree rules, precisely guides the positioning of core instances. This focused supervision significantly improves the handling of spatial ambiguities in referring expressions. Extensive testing on the ScanRefer benchmark shows that RG-SAN not only sets new performance standards, with a mIoU increase of 5.1 points, but also greatly enhances robustness in processing spatially ambiguous descriptions.

To sum up, our main contributions are as follows:

- We introduce RG-SAN, a novel approach for modeling spatial relationships among all entities in expressions, which enhances the model's referring ability in 3D-RES.

- We propose the TLM for precise localization of all instances mentioned in expressions, and RWS, utilizing only the target instance's location for supervising the spatial positioning of all instances.

- Extensive experiments on the ScanRefer benchmark demonstrate the effectiveness of the proposed RG-SAN, showing significant improvements in performance and robustness in 3D-RES tasks.

## 2 Related Work

### 2.1 3D Referring Expression Comprehension and Referring Expression Segmentation

Referring Expression Comprehension (REC) is proposed to locate the referred target from a short description of visual space by bounding boxes [74, 59, 29], which is part of vision-language tasks [12, 10, 11, 18, 69, 68, 15, 40, 41]. Recent works in 3D-REC can be divided into two parts, two-stage and single-stage. As for two-stage methods [5, 1, 75, 73, 72, 24, 13], 3D object proposals are generated directly from ground-truth [1] or extracted by a pre-trained 3D object detector [52] in the first stage,

and then assigned to language in the second stage. In the other way, some methods adopt a one-stage paradigm [47, 26, 70, 66], enabling end-to-end training.

Referring Expression Segmentation (RES) need fine-grained vision-language alignment [36, 37, 16, 35], proposed to locate the referred target by masks [27, 61, 25]. TGNN [24] introduce 3D-RES by extending the bounding box annotations of ScanRefer [5] to masks by incorporating the instance masks from ScanNet and proposed a two-stage pipeline. Further, 3D-STMN [65] proposed an end-to-end method that matches the text and superpoints to get the 3D segmentation of the target object directly.

## 2.2 3D Human-AI Interaction

ScanQA [3] has notably advanced visual question answering in 3D scenes, enhancing the human-AI interaction experience. Meanwhile, 3D-LLM [21], 3D-VisTA [77], NaviLLM [76], and BridgeQA [51] have further propelled this task. Li et al. [38, 39], Lu et al. [46] have explored how AI understands human instructions like gestures and language to locate targets. 3D-VisTA [77] introduced a new paradigm for large-scale 3D vision-language pre-training, greatly enhancing AI's understanding of 3D vision-language and advancing various downstream tasks. Works like 3D-LLM [21], Chat3D [64, 22], NaviLLM [76] and Scene-LLM [14] have extended the capabilities of multimodal large language models to the 3D realm, endowing embodied intelligence with the rich knowledge and capabilities of LLMs, thus ushering in the era of large models in Human-AI Interaction.

## 2.3 Weakly Supervision in Vision-and-Language

In the field of Vision Language, weakly supervised [33, 44, 34, 4] have gained significant attention and great progress. These approaches aim to tackle the challenge of limited or incomplete annotations by leveraging alternative supervised data or weakly labeled data. For weakly supervised visual question answering (VQA), Kervadec et al. [28] employ weak supervision in the form of object-word alignment as a pre-training task. Trott et al. [62] use object counts in images as weak supervision to guide VQA for counting-based questions. Gokhale et al. [17] employ logical connective rules to augment training datasets for yes-no questions. Weakly supervision from captions has also been employed for visual grounding tasks [9, 50, 2] recently. Especially, for RES, some methods [33, 44] localize the target object only using readily available image-text pairs.

# 3 Method

In this section, we provide a comprehensive overview of the RG-SAN. The framework is illustrated in Fig. 2. First, the features of visual and linguistic modalities are extracted in parallel (Sec. 3.1). Next, we demonstrate the process of TLM (Sec. 3.2.1). Finally, we outline the RWS and the training objectives (Sec. 3.3).

## 3.1 Feature Extraction

### 3.1.1 Visual Encoding

Given a point cloud scene $\mathbf{P}_{cloud} \in \mathbb{R}^{\mathcal{N}_p \times (3+F)}$ with $\mathcal{N}_p$ points. Each point comes with 3D coordinates along with an $F$-dimensional auxiliary feature that includes RGB, normal vectors, among others. We first employ a Sparse 3D U-Net [19] to extract point-wise features, represented as $\hat{\mathbf{P}}_{\mathbf{cloud}} \in \mathbb{R}^{\mathcal{N}_p \times C_p}$. Then, we follow Sun et al. [60] and Wu et al. [65] to obtain $\mathcal{N}_s$ superpoints $\{\mathcal{K}_i\}_{i=1}^{\mathcal{N}_s}$ [32] from the original point cloud. Finally, we directly feed point-wise features $\hat{\mathbf{P}}_{\mathbf{cloud}}$ into superpoint pooling layer based on $\{\mathcal{K}_i\}_{i=1}^{\mathcal{N}_s}$ to obtain the superpoint-level features $\mathbf{S}_p \in \mathbb{R}^{\mathcal{N}_s \times C_p}$.

### 3.1.2 Linguistic Encoding

Given a free-form plain text description of the target object, consisting of $\mathcal{N}_t$ words $\{c_i\}_{i=1}^{\mathcal{N}_t}$, we utilize a pre-trained MPNet model [58] to extract $C_t$-dimensional word-level embeddings, represented as $\mathbf{E}_0 \in \mathbb{R}^{\mathcal{N}_t \times C_t}$.

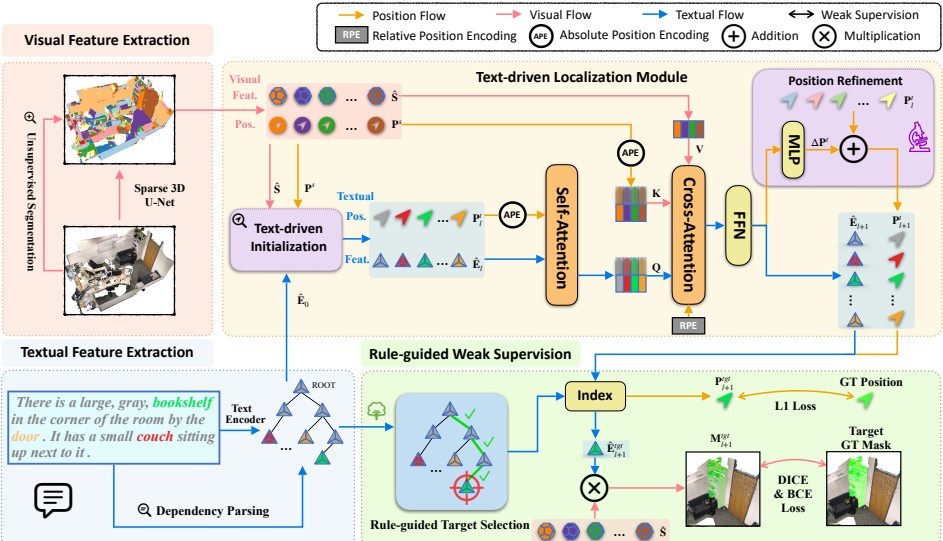

Figure 2: An overview of the proposed RG-SAN. This model analyzes a point cloud and a textual description with $\mathcal{N}_t$ tokens, extracting superpoints and word-level features. The TLM assigns spatial positions to tokens, facilitating multimodal fusion. The RWS strategy enables the model to learn the positions of all mentioned entities using only the supervision of the target position.

## 3.2 Context-driven Spatial Awareness

In this section, we address a key limitation in prior works that interact point clouds with text without considering spatial positioning [65, 47, 70]. Unlike these methods, which often lose spatial information due to unordered point cloud features, leading to ambiguous spatial relationship understanding, our approach is distinct. In 3D-RES, spatial information is inherently sparse and dynamic, depending on the specific target object described in the text, rather than the dense, static sampling of an entire point cloud scene [31].

To address this issue, we propose to facilitate interactions between textual entities and point clouds within 3D space, rather than merely at the semantic level. Specifically, our objective is to fully leverage semantic and spatial contextual information to accurately predict the spatial positions of all mentioned nouns within the point cloud.

Therefore, we introduce the Text-driven Localization Module (TLM) to initialize the positions of entity nouns in the text and continuously update and refine these positions through iterative multimodal interactions.

### 3.2.1 Text-driven Localization Module

Given the superpoint features $\mathbf{S}_p$ and word embeddings $\mathbf{E}_0$, we first project the features into the same dimension, and enhance the word-level embeddings by Dependency-Driven Interaction (DDI), following Wu et al. [65]:

$$\hat{\mathbf{E}}_0 = \mathbf{DDI}(\mathbf{E}_0 \mathbf{W}_{lang}), \quad \hat{\mathbf{S}} = \mathbf{S}_p \mathbf{W}_{vis}, \tag{1}$$

where $\mathbf{W}_{lang} \in \mathbb{R}^{C_t \times D}$ and $\mathbf{W}_{vis} \in \mathbb{R}^{C_p \times D}$ denote learnable parameters, and the subscript of $\mathbf{E}$ and $\hat{\mathbf{E}}$ represents the round number.

**Text-driven Initialization.** The key is to map the text into 3D geometric space in a meaningful way. Specifically, we enhance entity position prediction within point clouds through an interactive text-point cloud process. We do this by calculating feature similarity across modalities to accurately

estimate the spatial probability distribution for each mentioned entity:

$$\mathbf{E} = \hat{\mathbf{E}}_0 \mathbf{W}_E, \quad \mathbf{S} = \hat{\mathbf{S}} \mathbf{W}_S,$$
$$A_{ij} = \frac{\mathrm{Sim}(\mathbf{E}_i, \mathbf{S}_j)}{\sum_{j=1}^{\mathcal{N}_s} \mathrm{Sim}(\mathbf{E}_i, \mathbf{S}_j)}, \tag{2}$$

where $\hat{\mathbf{E}}_0$ denotes the initial word embeddings, $\hat{\mathbf{S}}$ denotes the superpoint features, $\mathbf{W}_E, \mathbf{W}_S \in \mathbb{R}^{D \times D}$ are learnable parameters, $A_{ij} \in \mathbb{R}$ denotes the probability of the $i$-th word token being located at the $j$-th superpoint, and $\mathrm{Sim}(\cdot, \cdot)$ represents the similarity function, which in this case is defined as $\mathrm{Sim}(\mathbf{E}, \mathbf{S}) = \exp(\mathbf{E}\mathbf{S}^T / \sqrt{D})$.

Following this, we utilize the spatial probability distribution $A$ to predict the approximate positions of the mentioned entities, as well as their corresponding representations:

$$\mathbf{P}_{0,i}^t = \sum_{j=1}^{\mathcal{N}_s} A_{ij} \mathbf{P}_j^s, \tag{3}$$

$$\mathbf{S}_v = \hat{\mathbf{S}} \mathbf{W}_v, \quad \hat{\mathbf{E}}_{0,i} = \sum_{j=1}^{\mathcal{N}_s} A_{ij} \mathbf{S}_{v,j}, \tag{4}$$

where $\mathbf{P}_j^s$ is the position of the $j$-th superpoint, $\mathbf{P}_{0,i}^t$ is the initial spatial position of $i$-th word token which will be refined iteratively as formulated in Sec. 3.2.2, $\mathbf{W}_v \in \mathbb{R}^{D \times D}$ denotes learnable parameters, and $\hat{\mathbf{E}}_{0,i}$ denotes the updated representation of the $i$-th word token. The sharing of distribution $A$ during centroid computation allows the entity representations to benefit from the guidance provided by spatial information, leading to a more accurate understanding of the 3D spatial relationships. Subsequently, the text and point clouds undergo multiple rounds of multimodal interactions, continually updating the embeddings and positions of the entities.

**Iterative Position Refinement.** After $l$-round multimodal interactions, the word tokens $\hat{\mathbf{E}}_l$, referred to as textual segment kernels, become increasingly precise, theoretically resulting in more accurate position predictions. A straightforward approach would involve replicating the initial interaction method by regressing position information in each round. However, following the methodologies of Redmon et al. [56] and Lai et al. [31], rather than directly optimizing the final position, we adopt a more manageable strategy of iteratively learning offsets. To this end, we refine the positions of textual tokens based on the evolving textual segment kernels. As depicted in Fig. 2, we employ a Multilayer Perceptron (MLP) to predict a position offset $\Delta \mathbf{P}_l^t = \mathbf{MLP}(\hat{\mathbf{E}}_{l+1}) \in \mathbb{R}^{\mathcal{N}_t \times 3}$ from the updated textual segment kernels $\hat{\mathbf{E}}_{l+1}$. This offset is then added to the previous textual positions $\mathbf{P}_l^t$:

$$\mathbf{P}_{l+1}^t = \mathbf{P}_l^t + \Delta \mathbf{P}_l^t. \tag{5}$$

This method allows for gradual refinement of position predictions, making the optimization process more effective and leading to progressively more accurate positioning with each iteration.

### 3.2.2 Spatial Awareness Aggregation

Once the positions of noun entities are obtained, techniques like positional encoding [63, 67, 31, 30, 6] can be used to further refine the positions.

**Absolute Positional Encoding (APE).** To initiate, we follow the approach of the original transformer [63] to encoded the positions of both superpoints and text tokens to obtain positional encodings $\mathbf{B}_l^s \in \mathbb{R}^{\mathcal{N}_s \times D}$ and $\mathbf{B}_l^t \in \mathbb{R}^{\mathcal{N}_t \times D}$ using absolute positional encoding (APE):

$$\mathbf{B}_l^s = \mathbf{APE}(\mathbf{P}_l^s), \quad \mathbf{B}_l^t = \mathbf{APE}(\mathbf{P}_l^t). \tag{6}$$

These positional encodings facilitate spatial-aware self-attention in the textural segment kernels $\hat{\mathbf{E}}_l$:

$$\dot{\mathbf{E}}_l = \mathbf{Attention}(\hat{\mathbf{E}}_l + \mathbf{B}_l^t, \hat{\mathbf{E}}_l + \mathbf{B}_l^t, \hat{\mathbf{E}}_l), \tag{7}$$

where Attention$(\cdot)$ uses the technique of Vaswani et al. [63] and $\mathbf{B}_l^t$ denotes the absolute positional encoding of $\hat{\mathbf{E}}_l$.

Next, we enhance textual and superpoint features with absolute positional encoding, and use them as Queries and Keys for subsequent multimodal aggregation:

$$\begin{aligned} \mathbf{Q} &= \text{Concat}(\dot{\mathbf{E}}_l, \mathbf{B}_l^t)\mathbf{W}_{query}, \\ \mathbf{K} &= \text{Concat}(\hat{\mathbf{S}}, \mathbf{B}_l^s)\mathbf{W}_{key}, \end{aligned} \tag{8}$$

where $\mathbf{B}_l^t \in \mathbb{R}^{\mathcal{N}_t \times D}, \mathbf{B}_l^s \in \mathbb{R}^{\mathcal{N}_s \times D}$ denote the absolute positional encoding of segmentation kernels and superpoints, respectively, and $\mathbf{W}_{query}, \mathbf{W}_{key} \in \mathbb{R}^{2D \times 2D}$ denote learnable parameters.

**Relative positional encoding (RPE).** For the further interaction with superpoint features, we adopt well-established relative positional encoding techniques [67, 31, 30, 6], such as Table-based RPE [67, 31] and 5D Euclidean RPE [6], which are formalized as follows:

$$\mathbf{B}_l^r[i, j] = \text{RPE}(\mathbf{Q}[i] + \mathbf{K}[j]), \tag{9}$$

where $\mathbf{B}_l^r[i, j] \in \mathbb{R}$ denotes the relative positional bias of the $i$-th $\mathbf{Q}$ relative to the $j$-th $\mathbf{K}$, $\text{RPE}(\cdot)$ denotes the operation of relative positional bias and $[\cdot]$ denotes the indexing operation.

Thus, we can perform multimodal aggregation enhanced with relative positional encoding:

$$\hat{\mathbf{E}}_{l+1} = \text{softmax}\left(\frac{\mathbf{Q} \cdot \mathbf{K}^T}{\sqrt{D}} + \mathbf{B}_l^r\right) \cdot (\hat{\mathbf{S}}\mathbf{W}_{val}), \tag{10}$$

where $\mathbf{W}_{val} \in \mathbb{R}^{D \times D}$ denote learnable parameters, $\mathbf{B}_l^r \in \mathbb{R}^{\mathcal{N}_t \times \mathcal{N}_s}$ denotes the relative positional bias, and $\hat{\mathbf{E}}_{l+1}$ denotes the updated segmentation kernels. This methodology significantly enriches the interaction between linguistic and 3D visual data, enabling more nuanced spatial understanding in our model.

### 3.3 Rule-guided Weak Supervision

### 3.3.1 Rule-guided Target Selection

In the preceding sections, we initially predicted the locations of all entities mentioned in the text. Ideally, supervised training would require position labels for each entity. However, we only have access to the location information of the target instance. This constraint leads us to adopt a weak supervision approach, focusing solely on the position of the referring instance for training. This approach introduces a significant challenge: accurately identifying the referring instance among the mentioned nouns. To address this, we utilize a pre-processed dependency tree, as outlined in Manning et al. [48], to accurately pinpoint the core noun, typically the subject of the sentence. We have developed a set of manual rules, based on this more general dependency tree, to enhance the identification process. These rules are specifically designed to guide the accurate positioning of core instances. The implementation of these rules is outlined in Algorithm 1.

---

**Algorithm 1** Rule-guided Target Selection

---

**Input:** The dependency tree $\mathcal{G} = (\mathcal{V}, \mathcal{E})$ of the textual description, where $\mathcal{V} = \{\text{token}\}$ denotes the set of nodes, $\mathcal{E} = \{(\text{relation, head, tail})\}$ denotes the set of relations between nodes.
**Output:** The index $i$ of Target Instance node $\mathcal{V}^{tgt}$
 1: Initialization $i$ to the root: $i = 0$
 2: find $\mathcal{E}_i$ with $\mathcal{V}_i$ as its head
 3: **if** $(\mathcal{E}_i \in \{\text{nsubj, compound}\})$ & $(\mathcal{V}_i \notin \{\text{which, that}\})$ **then**
 4:     $i \leftarrow \mathcal{E}_i$'s tail index
 5: **end if**
 6: **if** $\mathcal{V}_i \in \{\text{there, this, it, object}\}$ **then**
 7:     find $\mathcal{E}_i$ with $\mathcal{V}_i$ as its head
 8:     $i \leftarrow \mathcal{E}_i$'s tail index
 9: **end if**
10: **if** $\mathcal{V}_i \in \{\text{set, sets, color, shape}\}$ **then**
11:     find the first $\mathcal{E}_i$'s relation $\in \{\text{compound, nmod, dep}\}$
12:     $i \leftarrow \mathcal{E}_i$'s head index
13: **end if**

---

### 3.3.2 Training Objectives

Given the index of the target instance, we can directly obtain the corresponding segment kernel $\hat{\mathbf{E}}_{l+1}^{tgt} \in \mathbb{R}^D$ and position $\mathbf{P}_{l+1}^{tgt}$, which are then supervised by the target ground truth.

Then we perform matrix multiplication between $\hat{\mathbf{E}}_{l+1}^{tgt}$ and $\hat{\mathbf{S}}$ to get the predicted instance response maps, which can be formulated as

$$\mathbf{M}_{l+1} = \sigma(\hat{\mathbf{E}}_{l+1}^{tgt} \cdot \hat{\mathbf{S}}^T), \tag{11}$$

$$\mathbf{Mask}_{l+1} = \mathbf{M}_{l+1} > 0.5, \tag{12}$$

where $\mathbf{M}_{l+1} \in \mathbb{R}^{\mathcal{N}_s}$, $\mathbf{Mask}_{l+1} \in \{0,1\}^{\mathcal{N}_s}$ are the predicted response map and the instance mask corresponding to the target.

Given ground-truth binary mask of the referring expression $\mathbf{Y} \in \{0,1\}^{\mathcal{N}_p}$, we get the corresponding superpoint mask $\mathbf{Y}^s \in \{0,1\}^{\mathcal{N}_s}$ by superpoint pooling followed by a 0.5-threshold binarization, and then we apply the binary cross-entropy (BCE) loss on the final response map $\mathbf{M}_{l+1}$ following Sun et al. [60]. The operation can be written as:

$$\mathbf{Y}_i^s = \mathbb{I}(\sigma(\text{AvgPool}(\mathbf{Y}, \mathcal{K}_i))), \tag{13}$$

$$\mathcal{L}_{bce} = \text{BCE}(\mathbf{M}_{l+1}, \mathbf{Y}^s), \tag{14}$$

where $\text{AvgPool}(\cdot)$ denotes the superpoint average pooling operation, and $\mathbf{Y}_i^s$ denotes the binarized mask value of the $i$-th superpoint $\mathcal{K}_i$. $\mathbb{I}(\cdot)$ indicates whether the mask value is higher than 50%.

To tackle foreground-background sample imbalance, we can use Dice loss [49]:

$$\mathcal{L}_{dice} = \text{DICE}(\mathbf{M}_{l+1}, \mathbf{Y}^s). \tag{15}$$

To supervise the position $\mathbf{P}_{l+1}^{tgt}$, we use the center of the superpoints of the target instance $\mathbf{P}^{gt}$, as

$$\mathcal{L}_{pos} = \text{L1}(\mathbf{P}_{l+1}^{tgt}, \mathbf{P}^{gt}). \tag{16}$$

In addition, we add a simple auxiliary score loss $\mathcal{L}_{score}$ for mask quality prediction following Sun et al. [60].

Overall, the final training loss function $\mathcal{L}$ can be formulated as:

$$\begin{aligned} \mathcal{L} = \lambda_{bce}\mathcal{L}_{bce} + \lambda_{dice}\mathcal{L}_{dice} + \lambda_{pos}\mathcal{L}_{pos} + \\ \lambda_{score}\mathcal{L}_{score}, \end{aligned} \tag{17}$$

where $\lambda_{bce}$, $\lambda_{dice}$, $\lambda_{rel}$ and $\lambda_{score}$ are hyperparameters used to balance these four losses.

## 4 Expriment

### 4.1 Experiment Settings

In our experiment, we utilize the pre-trained Sparse 3D U-Net method to extract point-wise features from point clouds [60]. We also employ the pre-trained MPNet model [58] as our text encoder. For the rest of the network, training is conducted from scratch. We set an initial learning rate of 0.0001 and apply a learning rate decay at epochs 26, 34, and 46, each with a decay rate of 0.5. Our experiments use a default of 6 multiple rounds $L$, a batch size of 32, and a maximum sentence length of 80. We set $\lambda_{bce} = \lambda_{dice} = 1, \lambda_{pos} = \lambda_{score} = 0.5$. All experiments are conducted using PyTorch on a single NVIDIA Tesla A100 GPU, ensuring consistency in our computational process.

### 4.2 Dataset and Evaluation Metrics

We evaluate our method using the ScanRefer dataset, a recent 3D referring dataset [5, 24], comprising 51,583 English natural language expressions referring to 11,046 objects across 800 ScanNet scenes [7]. Following Chen et al. [5], our evaluation metrics include mean Intersection over Union (mIoU) and Acc@$k$IoU. "Unique" refers to cases where the target instance is the only one of its class, and "Multiple" indicates situations where there is at least one more object of the target's class.

Table 1: The 3D-RES results on ScanRefer. † The mIoU and accuracy are reevaluated on our machine. *We reproduce results by extracting points within the boxes as segmentation mask predictions using their official codes.

| Method | Unique (~19%) | | | Multiple (~81%) | | | Overall | | | Inference Time | | |
|---|---|---|---|---|---|---|---|---|---|---|---|---|
| | 0.25 | 0.5 | mIoU | 0.25 | 0.5 | mIoU | **0.25** | **0.5** | **mIoU** | Stage-1 | Stage-2 | All |
| Multi-task | | | | | | | | | | | | |
| EDA-box2mask [70] | 84.7 | 56.9 | - | 50.0 | 37.0 | - | 55.2 | 40.0 | 35.0 | - | - | - |
| 3DRefTR-SP [43] | 87.9 | 69.8 | - | 51.6 | 41.9 | - | 57.0 | 46.1 | 40.8 | - | - | 388ms |
| 3DRefTR-HR [43] | 89.6 | 77.0 | - | 52.3 | 43.7 | - | 57.9 | 48.7 | 41.2 | - | - | 405ms |
| UniSeg3D [71] | - | - | - | - | - | - | - | - | 29.6 | - | - | - |
| SegPoint [20] | - | - | - | - | - | - | - | - | 41.7 | - | - | - |
| Reason3D [23] | 88.4 | 84.2 | 74.6 | 50.5 | 31.7 | 34.1 | 57.9 | 41.9 | 42.0 | - | - | - |
| Single-task | | | | | | | | | | | | |
| TGNN [24] | - | - | - | - | - | - | 37.5 | 31.4 | 27.8 | - | - | - |
| TGNN† [24] | 69.3 | 57.8 | 50.7 | 31.2 | 26.6 | 23.6 | 38.6 | 32.7 | 28.8 | 26862ms | 235ms | 27097ms |
| InstanceRefer† [73] | 81.6 | 72.2 | 60.4 | 29.4 | 23.5 | 21.5 | 40.2 | 33.5 | 30.6 | 509ms | 672ms | 1181ms |
| X-RefSeg3D [54] | - | - | - | - | - | - | 40.3 | 33.8 | 29.9 | - | - | - |
| 3DVG-Transformer* [75] | 79.5 | 58.0 | 49.9 | 42.0 | 30.8 | 27.0 | 49.3 | 36.1 | 31.4 | - | - | - |
| 3D-SPS* [47] | 84.8 | 65.6 | 54.7 | 41.7 | 30.8 | 26.7 | 50.1 | 37.6 | 32.1 | - | - | - |
| 3DRESTR [43] | 79.0 | 54.2 | - | 40.2 | 22.1 | - | 46.0 | 26.9 | 28.7 | - | - | - |
| 3D-STMN [65] | **89.3** | 84.0 | **74.5** | 46.2 | 29.2 | 31.1 | 54.6 | 39.8 | 39.5 | - | - | 283ms |
| RG-SAN (Ours) | 89.2 | **84.3** | **74.5** | **55.0** | **35.4** | **37.4** | **61.7** | **44.9** | **44.6** | - | - | 295ms |

## 4.3 Quantitative Comparison

In our experiments on the ScanRefer dataset, our proposed RG-SAN demonstrates significant improvements in nearly all metrics on the single-task leaderboard, as shown in Tab. 1. Notably, RG-SAN shows substantial gains compared to the state-of-the-art single-task model 3D-STMN, with increases of 5.1 points in mIoU and 7.1 points in Acc@0.25. This highlights our model's inferencing capability. A more detailed examination reveals that the majority of these improvements occur in scenarios with multiple disruptive instances, where RG-SAN achieves a remarkable 6.3-point increase in mIoU. This setting, where the target instance is among other instances of the same type, demands discriminative reasoning from the model. The significant performance validates the enhanced referring capabilities empowered by spatial reasoning. Our proposed RG-SAN also outperforms multi-task models [70, 43], including LLM-based models [20, 23], in most 3D-RES metrics, despite those models benefiting from more annotated data.

Moreover, RG-SAN has competitive inference costs, being only 12ms slower than the efficient 3D-STMN and faster than all other compared models, demonstrating its high performance with minimal computational increase.

## 4.4 Ablation Study

### 4.4.1 Text-driven Localization Module

We conduct an ablation study on the Text-driven Localization Module (TLM), as illustrated in Tab. 2. Simultaneously, we perform a fine-grained analysis of various initialization schemes for embeddings and positions. The term "w/o TLM" denotes the approach of not modeling positional information and instead directly using text embeddings for interaction. "MAFT" refers to the direct adaptation of the method proposed in [31]. The "Project" method involves initializing embeddings based on text-driven embeddings and then projecting each textual token directly into a 3D position, while the "Random" method randomly assigns a position to each textual token. Finally, we utilize the initialization technique called Text-driven Initialization (TI), which simultaneously initializes both embeddings and positions in a text-driven manner. Tab. 2 clearly shows that, under identical conditions, TI outperforms the others in all metrics. This indicates that TI more effectively leverages positional information from the visual scene, leading to more precise initial positions for the textual tokens. Consequently, this reduces the complexity of the subsequent iterative refinement process, thereby enhancing the overall accuracy of our model in spatially aligning text with point cloud data. Additionally, Tab. 2 demonstrates that proper initialization leads to the superior performance of TLM compared to the methods without TLM.

Table 2: Ablation study of Text-driven Localization Module (TLM), where "w/o TLM" means not using TLM.

| Method | Init. of Embeddings | Init. of Positions | Multiple mIoU | Overall mIoU |
|---|---|---|---|---|
| w/o TLM | Text-driven | - | 32.5 | 40.3 |
| MAFT [31] | Zero | Random | 29.7 | 37.9 |
| Project | Text-driven | Project | 30.3 | 38.8 |
| Random | Text-driven | Random | 30.1 | 38.8 |
| TI | Text-driven | Text-driven | **34.1** | **42.3** |

Table 3: Ablation study of positional encoding, where "w/o Pos. Supervision" means not supervising the positions, and "w/o PE" means not using any positional encoding.

| positional encoding | Multiple | | | Overall | | |
|---|---|---|---|---|---|---|
| | 0.25 | 0.5 | mIoU | 0.25 | 0.5 | mIoU |
| w/o Pos. Supervision | 45.4 | 27.3 | 30.4 | 54.4 | 38.2 | 38.9 |
| w/o PE | 46.1 | 31.7 | 32.8 | 54.6 | 42.4 | 41.1 |
| Fourier APE | 46.0 | 30.9 | 32.0 | 55.1 | 41.5 | 40.7 |
| 5D Euclidean RPE | 46.7 | 32.5 | 33.3 | 54.6 | 43.9 | 41.7 |
| Table-based RPE | **47.2** | **33.7** | **34.1** | **55.6** | **43.9** | **42.3** |

#### 4.4.2 Positional Encoding

We compare various positional encoding methods previously employed in [57, 6, 31]. These methods include Fourier Absolute positional encoding (APE), 5D Euclidean Relative positional encoding (5D Euclidean RPE) [6], and Table-based Relative positional encoding (Table-based RPE) [31]. Tab. 3 reveals that Table-based RPE surpasses the other methods, suggesting that combining semantic information with relative relationships is advantageous. Additionally, we observe that employing only absolute positional encoding can result in lower performance than not using any positional encoding at all. This may be attributed to the inherent limitations of absolute positional encoding in capturing relative positional information. By complicating the semantic features, it introduces challenges in the model's training process, underscoring the importance of choosing the right positional encoding technique for effective performance.

#### 4.4.3 Rule-guided Weak Supervision

We conducted experiments employing various weakly supervised text kernel selection strategies to evaluate their efficacy in leveraging target annotations. The strategy labeled as "w/o RWS" involves selecting the token based on attention weight within the cross-attention module [65], while "Root" entails selecting the root token of the dependency tree. Table 4 illustrates that utilizing the root node as supervision slightly outperforms the "w/o RWS" baseline. This is likely due to the root node providing consistent supervision, whereas Top1 tends to select different nodes variably, which complicates the training process. In contrast, our Rule-guided Target Selection (RTS) strategy, based on dependency tree rules to locate subjects, aligns more effectively with the structural nature of the text. It precisely identifies the target entity's position, significantly enhancing annotation utilization and effectively directing model training. This leads to a notable improvement in model performance.

Furthermore, we conduct an ablation study on the impact of the position loss weight $\mathcal{L}_{pos}$, detailed in Tab. 5. We observe that increasing the weight generally improves performance, peaking at a weight of 0.5, beyond which performance begins to taper off. This finding highlights the importance of balancing the weight of the position loss to optimize the model's effectiveness.

#### 4.4.4 Comparison with MAFT

MAFT [31] has played a pivotal role in 3D instance segmentation by incorporating spatial position modeling, offering valuable insights into how spatial information can improve model performance. Inspired by this approach, we extend spatial information into the text space to better align visual and textual semantics, specifically targeting spatial relationship reasoning in 3D-RES. Our approach introduces two key innovations that distinguish it from MAFT:

- Unlike MAFT [31], which initializes queries with zeros and uses random initialization for positional information, we employ text-driven queries and positional information to model the spatial relationships of entities in the expressions. This allows our model to capture the spatial context better, resulting in a 4.4-point improvement in mIoU, as shown in Tab. 2

- In contrast to [31], which supervises the positions of all target instances, 3D-RES supervises only the core target word. Our novel RWS method constructs spatial relationships for all noun instances using only the target word's positional information, resulting in a 2.3-point improvement in mIoU, as demonstrated in Tab. 4.

Table 4: Weak Supervision Strategy in RWS, where "w/o RWS" means using attention-based Top1 approach in [65] instead of our RWS, and "RTS" refers to our Rule-guided Target Selection strategy.

| Strategy | Multiple | | | Overall | | |
|---|---|---|---|---|---|---|
| | 0.25 | 0.5 | mIoU | 0.25 | 0.5 | mIoU |
| w/o RWS | 47.2 | 33.7 | 34.1 | 55.6 | 43.9 | 42.3 |
| Root | 53.5 | 30.4 | 34.7 | 60.7 | 40.9 | 42.5 |
| RTS | **55.0** | **35.4** | **37.4** | **61.7** | **44.9** | **44.6** |

Table 5: Ablation study of the weight of $\mathcal{L}_{pos}$.

| Weight of $\mathcal{L}_{pos}$ | Multiple | | | Overall | | |
|---|---|---|---|---|---|---|
| | 0.25 | 0.5 | mIoU | 0.25 | 0.5 | mIoU |
| 0.1 | 54.7 | 34.9 | 36.9 | 61.3 | 44.3 | 44.0 |
| 0.2 | **55.5** | 34.0 | 37.0 | **62.0** | 43.7 | 44.2 |
| 0.5 | 55.0 | **35.4** | **37.4** | 61.7 | **44.9** | **44.6** |
| 1.0 | 55.3 | 34.5 | 37.0 | 61.8 | 44.2 | 44.3 |
| 2.0 | 54.3 | 33.8 | 36.6 | 61.0 | 43.7 | 43.9 |

| Description | Original Scene | Ground Truth | 3D-STMN | RG-SAN | Predicted Instance Locations |
|---|---|---|---|---|---|
| **(a)** This is a rectangular **tv**. It is above a small thin **table**. | | | | | [tv] [0.75, 2.94, 0.77]  [table] [0.64, 2.82, -0.07] |
| **(b)** There is a large, gray **bookshelf** in the corner of the room by the **door**. It has a small **couch** sitting up next to it. | | | | | [bookshelf] [-2.35, 0.78, 0.39]  [door] [-1.77, 1.66, 0.17]  [couch] [-1.57, 0.14, -0.53] |
| **(c)** There is a square brown **chair**. It is the one with a **coat** on it. | | | | | [chair] [0.88, -0.98, -0.54]  [coat] [0.79, -0.94, -0.46] |

Figure 3: Visualization of all the nouns in the textual description. Our RG-SAN can segment instances corresponding to different nouns, while 3D-STMN indiscriminately assigns all nouns to the target instance. **Zoom in for best view.**

## 4.5 Qualitative Comparison

We conduct a qualitative analysis on the ScanRefer validation set as shown in Fig. 3, comparing our proposed RG-SAN with 3D-STMN [65] to highlight our model's exceptional referring capability. Fig. 3 demonstrates our model's ability to accurately segment not only the target objects but also other nouns mentioned in the text. Unlike 3D-STMN, which misattributes all nouns to a single target, RG-SAN distinctly recognizes and locates each noun. For example, in Fig. 3-**(c)**, our model successfully identifies the target chair through relative positioning, even with similar objects in the scene, and accurately recognizes a coat as a supporting element in the description. This ability extends to Fig. 3-**(a)** and **(b)**, where RG-SAN correctly segments multiple auxiliary nouns into their corresponding instances, demonstrating its robust generalization for complex texts and precise localization for multiple entities. Such capabilities enhance the model's understanding of complex semantic scenes, significantly improving its ability to refer to specific entities accurately.

## 5 Conclusion

In this paper, we present RG-SAN to overcome the limitations of traditional 3D-RES methods, particularly their lack of spatial awareness. Specifically, the TLM is introduced to model and refine positional information, while the RWS is designed to employ dependency tree rules to accurately guide the position of the target object. Combining TLM with RWS strategy, RG-SAN significantly improves segmentation accuracy and robustly handles spatial ambiguities. Extensive experiments conducted on the ScanRefer benchmark demonstrate the superior performance of RG-SAN. This underscores the importance of incorporating spatial awareness into segmentation models, paving the way for future advancements in the domain.

# 6  Acknowledge

This work was supported by National Science and Technology Major Project (No. 2022ZD0118201), the National Science Fund for Distinguished Young Scholars (No.62025603), the National Natural Science Foundation of China (No. U22B2051, No. U21B2037, No. 62072389, No. 62302411, No. 623B2088), the Natural Science Foundation of Fujian Province of China (No.2021J06003) and China Postdoctoral Science Foundation (No. 2023M732948).

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

# Appendix

## A The Critical Role of Spatial Information in 3D-RES Tasks

Our analysis underscores the pivotal role spatial relations play in 3D-RES tasks. We assessed the ScanRefer dataset's referring expressions, classifying examples into two categories: those with spatial relation terms (e.g., "left", "right", "next", "bottom" and "side") as spatially related, and those without as spatially unrelated. Our findings revealed that spatially related samples form about 92% of the dataset, highlighting the prevalence of spatial descriptors. Additionally, the Sr3D dataset consists entirely of spatially related descriptions, and in the Nr3D dataset, a significant 90.5% of entries utilize spatial prepositions [1]. This evidence demonstrates the necessity of spatial descriptions in accurately identifying objects within a scene through natural language, emphasizing the essential need for effective spatial relation modeling in 3D-RES tasks.

## B 3D-RES on ReferIt3D Dataset

We extended the 3D-RES task on the ReferIt3D dataset [1] (which is also in English) by integrating instance masks from ScanNet and conducting relevant experiments, as shown in Tab. 6. In contrast to the original setup of ReferIt3D, we refrained from using ground truth bounding boxes or masks as input during our experiments, which significantly increased the level of difficulty. Nonetheless, our model achieved remarkable Acc@50 gains of **5.3** points for Sr3D and **2.9** points for Nr3D, accompanied by mIoU gains of 5.2 points for Sr3D and 1.0 points for Nr3D.

It is worth highlighting that our results demonstrate exceptional performance in terms of Acc@50 and mIoU. This can be attributed to the incorporation of spatial information, which enhances the accuracy of segmentation results and addresses the challenges of over-segmentation and under-segmentation encountered in previous approaches.

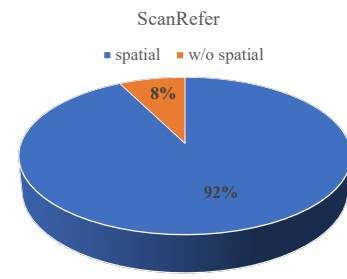

Figure 4: Statistics of samples in the ScanRefer dataset based on the presence of spatial relation descriptions, where "spatial" represents samples with spatially related descriptions, while "w/o spatial" denotes spatially unrelated samples.

## C More Ablation Studies

### C.1 Number of Multiple Rounds

We investigated the impact of varying the number of TLM rounds in our model. Analyzing rows two to five in Tab. 7 reveals a consistent pattern: performance improves with more rounds, reaches its peak at six, and then slightly declines. Fewer layers result in insufficient capacity, while an excessive number of layers increases the risk of overfitting. Therefore, selecting six layers strikes a balance that yields the best model performance.

In addition, we conducted ablation experiments to remove the iterative position refinement process at each layer. The results, shown in the first row of Tab. 7, clearly demonstrate the effectiveness of iterative refinement, leading to a significant improvement.

### C.2 The Textual Backbone

In Tab. 8, we compare the effects of commonly used natural language encoders. It can be observed that our method demonstrates robustness with respect to the selection of the NLP backbone. And we achieve the best performance using MPNet [58]. The underperformance of CLIP [55]is understandable, considering its optimization over a large dataset of text-image pairs. While CLIP excels at extracting representations at the sentence level, it encounters difficulties in comprehending intricate

Table 6: Results of 3D-RES tasks on ReferIt3D.

| Method | easy | | | hard | | | View Dep | | | View Indep | | | Overall | | | Inference Time |
|---|---|---|---|---|---|---|---|---|---|---|---|---|---|---|---|---|
| | 0.25 | 0.5 | mIoU | 0.25 | 0.5 | mIoU | 0.25 | 0.5 | mIoU | 0.25 | 0.5 | mIoU | 0.25 | 0.5 | mIoU | |
| **Nr3D** | | | | | | | | | | | | | | | | |
| TGNN | 29.2 | 22.3 | 21.0 | 22.5 | 19.6 | 17.4 | 22.2 | 18.1 | 17.0 | 27.6 | 22.4 | 20.3 | 25.7 | 20.9 | 19.1 | 26599ms |
| 3D-STMN | **47.9** | 31.9 | 32.6 | **35.4** | 20.0 | 23.0 | **37.7** | 21.1 | 24.3 | **43.5** | 28.4 | 29.4 | **41.5** | 25.8 | 27.6 | 276ms |
| RG-SAN | 45.8 | **34.5** | **33.5** | 34.5 | **23.3** | **24.0** | 37.3 | **26.3** | **26.3** | 41.5 | **30.1** | **29.8** | 40.1 | **28.7** | **28.6** | 289ms |
| **Sr3D** | | | | | | | | | | | | | | | | |
| TGNN | 28.2 | 23.0 | 20.9 | 29.1 | 25.8 | 21.9 | 23.8 | 21.3 | 18.2 | 28.6 | 23.9 | 21.3 | 27.5 | 22.9 | 20.2 | 26674ms |
| 3D-STMN | 49.4 | 38.2 | 36.3 | 41.9 | 31.0 | 30.1 | **45.5** | 33.5 | 31.9 | 47.2 | 36.2 | 34.6 | 47.2 | 36.1 | 34.4 | 281ms |
| RG-SAN | **55.8** | **43.6** | **41.5** | **46.7** | **36.3** | **35.2** | 42.6 | **34.7** | **32.5** | **53.5** | **41.7** | **39.9** | **53.1** | **41.4** | **39.6** | 293ms |

Table 7: Number of multiple rounds.

| Iter. Refine | Number of Rounds | Multiple | | | Overall | | |
|---|---|---|---|---|---|---|---|
| | | 0.25 | 0.5 | mIoU | **0.25** | **0.5** | **mIoU** |
| | 6 | 55.2 | 33.0 | 36.3 | 61.8 | 42.9 | 43.7 |
| ✓ | 1 | **56.3** | 30.3 | 35.6 | **62.8** | 40.6 | 43.1 |
| ✓ | 3 | 55.2 | 34.5 | 37.1 | 61.7 | 44.1 | 44.3 |
| ✓ | 6 | 55.0 | 35.4 | **37.4** | 61.7 | 44.9 | **44.6** |
| ✓ | 9 | 53.1 | **35.8** | 37.0 | 60.0 | **45.2** | 44.2 |

spatial relationships within sentences. This limitation hampers its performance in situations involving multiple objects, resulting in relatively poorer results.

## C.3 The Visual Backbone

We explored alternative visual backbones, including the PointNet++ [53] pretrained by the classic work 3D-VisTA [77] and another superpoint-based backbone, SSTNet [42], as detailed in Tab. 9. Our findings indicate that the performance with PointNet++, SSTNet and our employed SPFormer are quite comparable, demonstrating the adaptability and effectiveness of our proposed modules across different backbone architectures.

# D More Qualitative Analysis

More qualitative comparison results are illustrated in Fig. 5 and Fig. 6, demonstrating the remarkable discriminative ability of our RG-SAN compared to 3D-STMN. Fig. 5 showcases RG-SAN's superior performance in accurately localizing target objects, especially in challenging scenarios that require understanding complex positional relationships described in the text. For instance, Fig. 5-(**b**) illustrates a scenario with numerous distractors and a complex textual description, where 3D-STMN fails, causing over-segmentation. In contrast, RG-SAN accurately discerns and localizes the target object amidst distractions, achieving higher-quality segmentation. It is important to highlight that when faced with descriptive text that involves spatial relationship reasoning among multiple instances mentioned, as seen in all cases in Fig. 5, our RG-SAN demonstrates the capability to precisely locate and identify the target object. In contrast, 3D-STMN[65] lacks comparable complex reasoning abilities in such scenarios.

Table 8: Ablation study comparing text encoders.

| Text Encoder | Multiple | | | Overall | | |
|---|---|---|---|---|---|---|
| | 0.25 | 0.5 | mIoU | **0.25** | **0.5** | **mIoU** |
| BERT [8] | 53.2 | 34.8 | 36.4 | 60.0 | 44.1 | 43.7 |
| RoBERTa [45] | 53.1 | 34.8 | 36.7 | 60.0 | 44.0 | 44.0 |
| CLIP [55] | 52.3 | 33.4 | 35.0 | 58.2 | 42.1 | 42.5 |
| MPNet [58] | **55.0** | **35.4** | **37.4** | **61.7** | **44.9** | **44.6** |

Table 9: Ablation study comparing visual backbones.

| Text Encoder | Multiple | | | Overall | | |
|---|---|---|---|---|---|---|
| | 0.25 | 0.5 | mIoU | **0.25** | **0.5** | **mIoU** |
| SSTNet [8] | 53.7 | 34.3 | 34.9 | 59.4 | 42.5 | 43.2 |
| PointNet++ [45] | 54.1 | 34.6 | 36.1 | 60.3 | 44.2 | 44.0 |
| SPFormer [55] | **55.0** | **35.4** | **37.4** | **61.7** | **44.9** | **44.6** |

In Fig. 6, we visualized the predicted masks of the mentioned instances of our RG-SAN. As can be seen in Fig. 6 **(b)**, even if the "coat" category is not present in the training labels, our RG-SAN is still able to accurately identify the mentioned "coat" in the point cloud scene. This is because we align the word features from the textual modality and the point cloud features from the visual modality in a fine-grained manner through weak supervision. This alignment brings them into the same feature space, enabling the model to have strong generalization capabilities for unknown semantic categories. This paves the way for future research in weak supervision and open vocabulary.

Furthermore, as seen in Fig. 6 **(f)**, our RG-SAN is even able to accurately recognize the plural form of the entity noun "couches" mentioned in the descriptive text, while successfully identifying the target object. This capability enables the model to have a more precise and efficient understanding of spatial relationships associated with multiple auxiliary objects, such as "between" and "among", showcasing the powerful spatial relationship modeling ability of our RG-SAN.

## E  Analysis of Target Word Positioning Capability in LLMs

We attempt to utilize LLMs, specifically LLAMA 2 70B as an example, for target word positioning. To achieve this, we construct a command template for LLMs, which includes the input description token list and demonstration examples. Such a general template is designed as follows:

$$
\begin{aligned}
& \text{`` Given a word list, find the target word in the list:} \\
& \quad \text{['the', 'trash', 'can', 'is', 'directly', 'right',} \\
& \quad \text{'of', 'the', 'brown', 'tables', 'turned',} \\
& \quad \text{'sideways', '. '] => 'can', 2} \\
& \quad \text{['there', 'is', 'a', 'dark', 'brown', 'wooden',} \\
& \quad \text{'and', 'leather', 'chair', '. ', 'placed', 'in', 'the',} \\
& \quad \text{'table', 'of', 'the', 'kitchen', '. '] => 'chair', 8} \\
& \quad \text{[LIST]: => ''}
\end{aligned}
\tag{18}
$$

where [LIST] is replaced by the input description token list, and " => 'can', 2 " denotes the target token in the first example is "can" whose index in the token list is 2.

For the ScanRefer dataset, LLAMA 2 70B produces approximately 80% of target word positions that align with the results obtained from our RWS module. In the remaining portion, our RWS module demonstrates higher accuracy. This partially indicates that there is still room for improvement in LLAMA 2 70B's ability to identify target word positions. Conversely, our rule-based approach benefits from efficient utilization of explicit dependency relationships and exhibits certain advantages. Additionally, LLAMA 2 70B poses a significant computational burden. Taking this into consideration, our adopted RWS approach outperforms LLAMA 2 70B in terms of both accuracy and efficiency.

## F  Limitations and Broader Impact

Despite the strong performance of RG-SAN, we identify several limitations that call for further improvement. A primary limitation is its difficulty in accurately localizing plural nouns. This issue arises from the method of using a single point for localization, which proves challenging for plural entities in certain contexts. In future work, we will explore using multiple points to delineate boundaries for more precise localization of plural nouns.

The second limitation of our study concerns the model's inadequate robustness towards damaged point cloud data. The occurrence of damaged or incomplete data within point cloud datasets presents

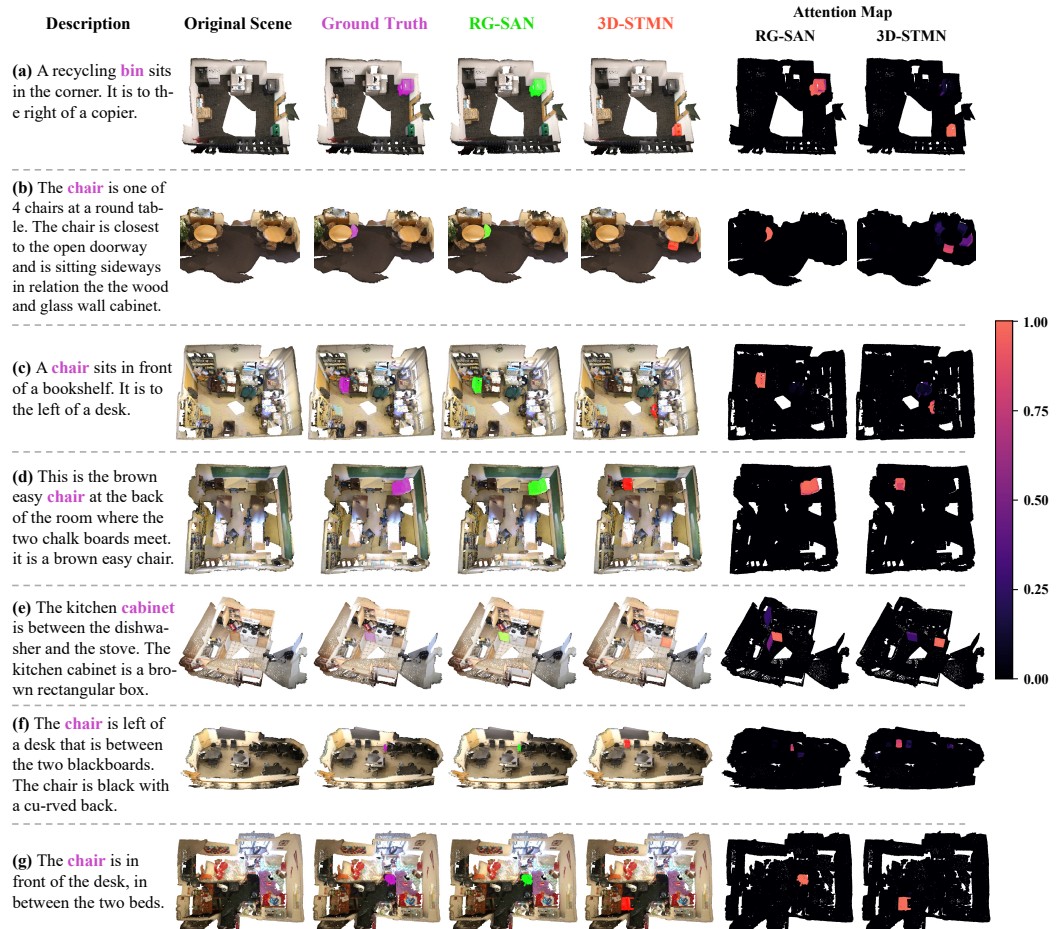

Figure 5: Qualitative comparison between the proposed RG-SAN and 3D-STMN. **Zoom in for best view.**

a significant challenge, one that our current model is not sufficiently equipped to address. This lack of robustness can impair the model's ability to process such data accurately, leading to unreliable results in scenarios involving incomplete or corrupted point clouds. Future work will aim to enhance the model's resilience and capability in handling and compensating for data imperfections.

RG-SAN is expected to stimulate further development and application of multimodal 3D perception, especially in practical scenarios such as embodied intelligence and autonomous driving. However, when it comes to practical applications, particularly those involving safety and privacy, rigorous testing is required to ensure compliance with relevant laws and regulations.

## G  Ethics Statement and Licenses

In our work, there are no human subjects and informed consent is not applicable. Additionally, we use publicly available text data from the ScanRefer Dataset (`https://daveredrum.github.io/ScanRefer`), which is licensed under a *Creative Commons Attribution-NonCommercial-ShareAlike 3.0 Unported License* which allows us to use the dataset for non-commercial purposes. For point cloud data, we used the publicly available ScanNet Dataset (`https://github.com/ScanNet/ScanNet`), which is licensed under the ScanNet Terms of Use, and the code is released under the MIT license. Both the licenses of ScanNet allow us to use the dataset and code for non-commercial purposes. In the appendix, we use the ReferIt3D Dataset (`https://github.com/referit3d/referit3d`) for extra experiments, which is licensed under the MIT license which allows us to use the dataset for non-commercial purposes.

| Description | Original Scene | Ground Truth | Predicted Masks of Mentioned Instances |

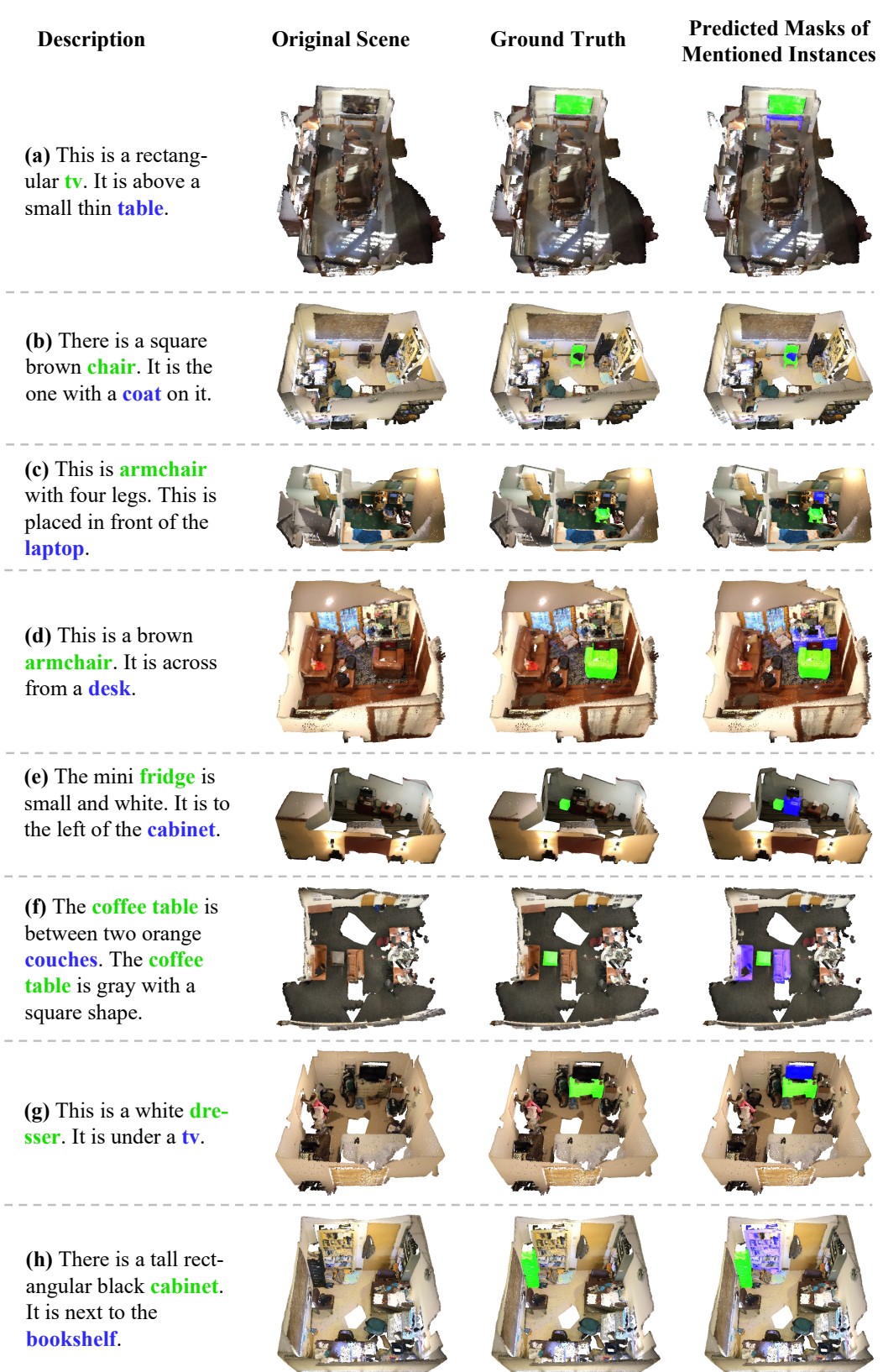

**(a)** This is a rectangular **tv**. It is above a small thin **table**.

**(b)** There is a square brown **chair**. It is the one with a **coat** on it.

**(c)** This is **armchair** with four legs. This is placed in front of the **laptop**.

**(d)** This is a brown **armchair**. It is across from a **desk**.

**(e)** The mini **fridge** is small and white. It is to the left of the **cabinet**.

**(f)** The **coffee table** is between two orange **couches**. The **coffee table** is gray with a square shape.

**(g)** This is a white **dresser**. It is under a **tv**.

**(h)** There is a tall rectangular black **cabinet**. It is next to the **bookshelf**.

Figure 6: The visualization of the predicted masks of mentioned instances of our RG-SAN. **Zoom in for best view.**

