# OpenReview forum: "RG-SAN: Rule-Guided Spatial Awareness Network for End-to-End 3D Referring Expression Segmentation"
_NeurIPS.cc/2024/Conference — NeurIPS 2024 oral_

### Official Review · Reviewer_MnE2 · 2024-07-05

**Soundness:** 3
**Presentation:** 4
**Contribution:** 4
**Rating:** 8
**Confidence:** 5

**Summary:**

This paper presents RG-SAN, a new method for 3D Referring Expression Segmentation. It combines spatial reasoning with textual cues to segment 3D objects accurately. RG-SAN uses a Text-driven Localization Module and a Rule-guided Weak Supervision strategy. It outperforms existing methods on the ScanRefer benchmark. It handles spatial ambiguities well and sets a new precision standard for 3D scene understanding. Extensive ablation studies and visualizations validate the effectiveness of the proposed modules.

**Strengths:**

1) The motivation of this paper is clear. Figure 1 shows how spatial relationship info in natural language matches with 3D scenes, making the motivation obvious. Figure 4 reinforces this motivation from a quantitative perspective through statistical analysis.

2) I agree that spatial relationship reasoning is crucial for understanding 3D scenes. The complex spatial relationships in natural language give important spatial clues. The proposed Text-driven Localization Module (TLM) performs spatial reasoning explicitly, aligning with how humans understand 3D scenes, which makes sense to me.

3) The Rule-guided Weak Supervision (RWS) module enables localization and segmentation of auxiliary object nouns in natural language without any supervision. This aspect is interesting and shows the model's generalization capability.

4) The appendix includes an analysis of how well large language models (LLMs) can localize target words, comparing this with the RWS module's results. This comparison further validates RWS and offers new insights into using LLMs for the 3D-RES task.

5) The paper reports extensive experiments on common 3D-RES datasets like ScanRefer and ReferIt3D, achieving state-of-the-art performance. Ablation studies also show the effectiveness of the TLM and RWS modules

**Weaknesses:**

1) The paper does a good job comparing 3D-RES methods. But traditional 3D Visual Grounding methods using bounding boxes are more mature. I'd like to see how these older methods perform on this task. This would give a more complete quantitative comparison with the proposed method.

2) The appendix talks about LLMs for target word localization. But it doesn't compare them directly with 3D Visual Grounding methods based on LLMs. Could LLM-based approaches be better? I'd like to see a comparison between specialized lightweight models and general LLMs.

3) The paper should provide details on the superpoint feature extraction mentioned in line 119. I'm curious if superpoint features cover all targets. If not, what's the missing rate?

4) The layout of Tables 2 and 3 is off, and the font in Figure 2 is too small. This makes it hard to read the details.

5) The text processing procedure isn't detailed enough. For example, the interaction process of the DDI module. I'd recommend including this description in the main text.

**Questions:**

1 Will the code for RG-SAN be open-sourced in the future?

2 Could the authors provide a quantitative comparison of RG-SAN with a broader range of 3D visual grounding methods?

Others please look at weaknesses.

**Limitations:**

Limitations are discussed [line 576].

---

> ### Author Rebuttal · Authors · 2024-08-06
>
> Thank you for your positive feedback and recognition of our contributions. We greatly appreciate your commendation of our clear articulation of motivation and quantitative analysis. We're pleased you acknowledged our method's effectiveness in spatial relation reasoning for 3D scenes, expressing interest in our rule-guided weak supervision strategy. Now, let's address the specific concerns you've raised and offer further clarification:
>
> ---
>
> > **Q1: The paper does a good job comparing 3D-RES methods. But traditional 3D Visual Grounding methods using bounding boxes are more mature. I'd like to see how these older methods perform on this task. This would give a more complete quantitative comparison with the proposed method.**
> >
>
> A1: Thank you for your insightful suggestion. Based on your advice, we adapted the high-performing methods 3DVG-Transformer and 3D-SPS from 3D-REC for 3D-RES and tested their performance, as shown in Table A. Our method still demonstrates a significant advantage, outperforming by over 10 points.
>
> | Method | Unique mIoU | Multiple mIoU | Overall mIoU |
> | --- | --- | --- | --- |
> | 3DVG-Transformer [a]* | 49.9 | 27.0 | 31.4 |
> | 3D-SPS [b]* | 54.7 | 26.7 | 32.1 |
> | RG-SAN (Ours) |  74.5 | 37.4 | 44.6 |
>
> Table A: Comparison with 3D Visual Grouding methods. * we reproduce results by extracting points within the boxes as segmentation mask predictions using their official codes.
>
> [a] 3DVG-Transformer: Relation modeling for visual grounding on point clouds. ICCV 2021
>
> [b] 3d-sps: Single-stage 3d visual grounding via referred point progressive selection. CVPR 2022
>
> ---
>
> > **Q2: The appendix talks about LLMs for target word localization. But it doesn't compare them directly with 3D Visual Grounding methods based on LLMs. Could LLM-based approaches be better? I'd like to see a comparison between specialized lightweight models and general LLMs.**
> >
>
> A2: Thank you for your valuable suggestion. We compared our model with LLM-based 3D RES models SegPoint [c] and Reason3D [d], as shown in Table B, and our model still demonstrates a significant advantage, leading by more than 2.6 points.
>
> | Method | Unique mIoU | Multiple mIoU | Overall mIoU |
> | --- | --- | --- | --- |
> | SegPoint [c] | - | - | 41.7 |
> | Reason3D [d] | 74.6 | 34.1 | 42.0 |
> | RG-SAN (ours) |  74.5 | 37.4 | 44.6 |
>
> Table B: Comparison with LLM-based methods.
>
> [c] SegPoint: Segment Any Point Cloud via Large Language Model. ECCV 2024
>
> [d] Reason3D: Searching and Reasoning 3D Segmentation via Large Language Model. Arxiv 2024
>
> ---
>
> > **Q3: The paper should provide details on the superpoint feature extraction mentioned in line 119. I'm curious if superpoint features cover all targets. If not, what's the missing rate?**
> >
>
> A3: We appreciate your insightful feedback. The superpoint generation mechanism in RG-SAN ensures that no instances are missed, as the superpoints comprehensively cover the entire scene. Superpoints are essentially fine-grained fragments that group semantically similar points together. They do not overlap with each other and collectively constitute the whole scene, guaranteeing that all objects are included within superpoints. This setup ensures that every object is included within the superpoints, as detailed in the upper left corner of Figure 2 in our paper. Therefore, the issue of missing objects does not arise in our framework.
>
> ---
>
> > **Q4: The layout of Tables 2 and 3 is off, and the font in Figure 2 is too small. This makes it hard to read the details.**
> >
>
> A4: Thank you for your suggestions. To improve readability and facilitate understanding, we will adjust the layout of Tables 2 and 3 and enhance Figure 2, including resizing the font, in the new version.
>
> ---
>
> > **Q5: The text processing procedure isn't detailed enough. For example, the interaction process of the DDI module. I'd recommend including this description in the main text.**
> >
>
> A5: Thank you for your suggestion. We will include the details of DDI interactions in the new version to enhance understanding for readers.
>
> ---
>
> > **Q6: Will the code for RG-SAN be open-sourced in the future?**
> >
>
> A6: Thanks for your interest. In the paper, we have already provided the code via an anonymous link and also uploaded a copy in the supplementary materials. We also commit to releasing the code once the paper is accepted.

---

> > ### Author Response · Authors · 2024-08-09
> > **Sincere Request for Further Discussions**
> >
> > Dear Reviewer MnE2,
> >
> > Thanks again for your great efforts and constructive advice in reviewing this paper! With the discussion period drawing to a close, we expect your feedback and thoughts on our reply. We put a significant effort into our response, with several new experiments and discussions. We sincerely hope you can consider our reply in your assessment.
> >
> > We look forward to hearing from you, and we can further address unclear explanations and remaining concerns if any.
> >
> > Regards, Authors

---

> > ### Comment · Reviewer_MnE2 · 2024-08-09
> > **Raise the score to 8**
> >
> > Thanks to the authors for an excellent rebuttal—I'm pleased to say that all of my concerns have been thoroughly addressed.
> >
> > The inclusion of both traditional visual grounding and the latest LLM-based approaches really strengthens the paper's conclusions. I'm particularly excited about the LLM-based approach, and I believe expanding on this in future versions could really push the field forward. It’s clear that this aspect has a lot of potential to guide future research.
> >
> > I also took the time to review the other reviewers’ comments, and I stand by my initial impression. The authors' exploration of how spatial relationships in natural language correspond with 3D scenes tackles a crucial and challenging area, especially compared to purely visual 3D segmentation. Spatial and relational reasoning is one of the major hurdles in cross-modal 3D vision today, and it’s great to see the authors making strides in this direction. I’m confident this work will inspire further progress in embodied intelligence. I will fully support this paper and raise its score.

---

> > > ### Author Response · Authors · 2024-08-14
> > >
> > > Thank you very much for your recognition. We will incorporate everyone’s feedback into the final version and make the code open source for the community to learn from. Once again, we sincerely appreciate your suggestions.

---

### Official Review · Reviewer_8gkL · 2024-07-06

**Soundness:** 4
**Presentation:** 4
**Contribution:** 4
**Rating:** 8
**Confidence:** 5

**Summary:**

This paper presents a novel and high-performing 3D referring segmentation network. Specifically, it approaches the problem from both 3D spatial relationships and natural language spatial descriptions, innovatively using explicit spatial position modeling and multimodal interaction. This allows the query corresponding to textual entities to understand both semantics and spatial locations. Additionally, the use of weak supervision techniques enables the model to achieve strong generalization capabilities even under incomplete annotations. Comprehensive experiments further validate the superior performance of the proposed method.

**Strengths:**

I commend the authors for their insightful paper, particularly the proposed text-guided spatial perception modeling approach. This method aligns with human cognitive habits and has the potential to significantly advance the field of multimodal 3D perception. Several notable advantages are highlighted:
1. The paper deeply explores text-conditioned 3D spatial perception from both 3D spatial relationships and natural language structure perspectives, advancing the community's exploration of multimodal spatial perception modeling.
2. The proposed TLM module effectively addresses the challenge of explicit spatial reasoning in previous end-to-end segmentation paradigms, significantly improving segmentation performance while maintaining high inference speed.
3. The RWS module demonstrates data efficiency, generalizing capabilities to all entities without requiring mask labels for all textual entities.
4. The experiments are comprehensive, evaluating the model's performance on both the ScanRefer and ReferIt3D datasets, thoroughly validating its robust performance.
5. The ablation studies are detailed, thoroughly analyzing the proposed TLM and RWS modules, as well as the backbone selection and hyperparameter settings.
6. The video demo in the open-source link is engaging, and the visualizations in Figure 3 of the paper are intuitive, effectively illustrating the core ideas and powerful performance of the proposed method.

**Weaknesses:**

1. More details can be added regarding the superpoint feature extraction and text feature processing sections.
2. The paper mentions that the Sparse 3D U-Net used as the visual backbone is pre-trained. On which datasets was it pre-trained? Would using different pre-trained backbones result in performance variations?
3. Figure 2 has too many colors, making it somewhat cluttered and potentially confusing for readers. It is recommended to simplify and optimize the color scheme.
4. It is suggested to include some bad cases to enhance the completeness of the work.

**Questions:**

Will the complete code for the paper be open-sourced for additional exploration?

**Limitations:**

The authors discuss the limitations in the appendix.

---

> ### Author Rebuttal · Authors · 2024-08-06
>
> Thank you for your positive feedback and recognition of our work. We appreciate your acknowledgment of our exploration of text-conditioned 3D spatial perception and the effectiveness of the TLM and RWS modules. We're also glad you found our video demo and Figure 3 visualizations clear and insightful. Now, let's address your specific concerns and provide further clarification:
>
> ---
>
> > **Q1: More details can be added regarding the superpoint feature extraction and text feature processing sections.**
> >
>
> A1: We appreciate your valuable suggestions. We will provide a detailed description of superpoint feature extraction and text feature processing in the new version.
>
> ---
>
> > **Q2: The paper mentions that the Sparse 3D U-Net used as the visual backbone is pre-trained. On which datasets was it pre-trained? Would using different pre-trained backbones result in performance variations?**
> >
>
> A2: Thank you for your insightful question. The 3D U-Net we used has been pre-trained on 3D instance segmentation tasks [45]. Additionally, following your suggestion, we explored alternative backbones, including PointNet++ [39], used by the classic work 3D-VisTA [a], and another superpoint-based backbone, SSTNet [28], as detailed in Table A. Our findings indicate that the performance with PointNet++ [45] and our employed SPFormer [45] are comparable, demonstrating the adaptability and effectiveness of our proposed modules across different backbone architectures. We will include this discussion in the final version.
>
> | Visual Backbone | Unique mIoU | Multiple mIoU | Overall mIoU |
> | --- | --- | --- | --- |
> | SSTNet [28] | 73.9 | 33.9 | 42.0 |
> | PointNet++ [39] | 75.5 | 36.1 | 44.0 |
> | SPformer [45] |  74.5 | 37.4 | 44.6 |
>
> Table A: Ablation study of the Visual Backbones.
>
> [a] 3D-VisTA: Pre-trained Transformer for 3D Vision and Text Alignment. ICCV 2023.
>
> ---
>
> > **Q3: Figure 2 has too many colors, making it somewhat cluttered and potentially confusing for readers. It is recommended to simplify and optimize the color scheme.**
> >
>
> A3: Thank you for your detailed feedback. We will optimize Figure 2 in the next version to enhance its clarity and readability.
>
> ---
>
> > **Q4: It is suggested to include some bad cases to enhance the completeness of the work.**
> >
>
> A4: Thank you for your valuable suggestions. We will include bad cases and corresponding analyses in the new version.
>
> ---
>
> > **Q5: Will the complete code for the paper be open-sourced for additional exploration?**
> >
>
> A5: Thank you for your interest. In the paper, we have already provided the code via an anonymous link and also uploaded a copy in the supplementary materials. We also commit to releasing the complete code once the paper is accepted.

---

> > ### Author Response · Authors · 2024-08-09
> > **Sincere Request for Further Discussions**
> >
> > Dear Reviewer 8gkL,
> >
> > Thanks again for your great efforts and constructive advice in reviewing this paper! With the discussion period drawing to a close, we expect your feedback and thoughts on our reply. We put a significant effort into our response, with several new experiments and discussions. We sincerely hope you can consider our reply in your assessment.
> >
> > We look forward to hearing from you, and we can further address unclear explanations and remaining concerns if any.
> >
> > Regards, Authors

---

> > ### Author Response · Authors · 2024-08-13
> >
> > Dear Reviewer 8gkL,
> >
> > We are grateful for your thorough review and the constructive feedback provided on our submission. Your insights have significantly contributed to the refinement of our paper. We have endeavored to address all the points raised in your initial review comprehensively.
> >
> > As the discussion period for NeurIPS 2024 is drawing to a close, we would appreciate knowing if there are any further clarifications or additional details you might need. We are fully prepared to continue discussions to further enhance the quality of our work.
> >
> > With appreciation,
> >
> > Paper 9950 Authors

---

> ### Comment · Reviewer_8gkL · 2024-08-14
>
> I apologize for the delayed response. I've been quite busy lately, but I wanted to take a moment to wrap things up. First, I’d like to thank the authors for their detailed response. It’s impressive to see that the proposed method performs effectively across different backbones. After carefully reviewing all the discussions, I find this paper to be very valuable. The exploration of text-conditioned 3D spatial perception from both 3D spatial relationships and natural language structure perspectives provides constructive guidance for 3D cross-modal understanding, which is indeed a challenging aspect of human-computer interaction. The authors have elegantly addressed this issue without introducing additional data or annotations, which is truly inspiring. The contributions of this paper have been widely recognized by everyone involved.
>
> I also noted Reviewer 6dEW's comments regarding some minor issues with the formula descriptions. In my view, these do not affect the overall readability of the paper and can be easily addressed with minor revisions. Therefore, I believe this paper deserves a strong score, and I am willing to champion it.

---

> > ### Author Response · Authors · 2024-08-14
> >
> > Thank you very much for recognizing our work on text-conditioned 3D spatial perception. We will incorporate your feedback into the final version and make the code open source for the community to learn from. Once again, we sincerely appreciate your suggestions.

---

### Official Review · Reviewer_p7dx · 2024-07-06

**Soundness:** 4
**Presentation:** 4
**Contribution:** 4
**Rating:** 7
**Confidence:** 5

**Summary:**

This paper presents the Rule-Guided Spatial Awareness Network (RG-SAN) for 3D referring expression segmentation (3D-RES), offering a novel approach to understanding spatial relationships in the visual-language perception domain. It aligns 3D and linguistic features not only at the semantic level but also within geometric space. The proposed network incorporates modules for textual feature extraction, text-driven localization, and rule-guided weak supervision. In the experimental setup, the model builds upon the efficient Superpoint Transformer for 3D feature extraction, as developed by Sun et al. (2023). The experimental results are promising, particularly in the overall 0.25 threshold setting on the ScanRefer dataset. Extensive ablation studies demonstrate the effectiveness of the proposed modules, while vivid visualizations showcase the method's impressive generalization capabilities.

**Strengths:**

1. RG-SAN approaches 3D-language semantic alignment and spatial perception from a novel perspective. It not only aligns language features with 3D point cloud features at the semantic level but also explicitly assigns spatial positions to textual entity words within the geometric space. This explicit alignment helps address the spatial ambiguity inherent in directional natural language, allowing RG-SAN to achieve a more precise understanding of spatial relationships described in text.
2. The authors conducted comprehensive experiments on the 3D-RES task, with particularly notable performance improvements on the ScanRefer dataset. It is impressive that the method significantly enhances performance while maintaining rapid inference speed, which is beneficial for real-time applications of this task.
3. The authors performed detailed ablation studies on the proposed TLM and RWS modules, thoroughly examining the settings and hyperparameter choices. Additionally, they conducted ablation studies on the visual backbone and text backbone in the supplementary materials. These comprehensive ablations help readers understand the efficacy and rationale behind the proposed modules.
4. In the supplementary materials, the authors provided a statistical analysis of the importance of spatial information for the 3D-RES task. This quantitative analysis supports the motivation of the paper and gives readers a clearer understanding of the role of spatial information in this task.
5. The authors' visualizations are illustrative. In particular, Figure 3 demonstrates RG-SAN's text-guided spatial understanding and localization capabilities, showcasing excellent generalization.
6. The authors have committed to open-sourcing their method, providing a link that includes the source code and an engaging video demo. This openness will promote development and knowledge sharing within the community.

**Weaknesses:**

1. RG-SAN adopts superpoints as fundamental visual units for feature extraction and segmentation. While previous works have employed similar approaches, I am interested in understanding the segmentation quality of superpoints themselves. For instance, how many superpoints exist solely within individual objects? This is crucial because if a superpoint spans across two objects, it inevitably affects the segmentation results.
2. RG-SAN trains spatial awareness networks using the centroid coordinates of target objects. Here, does "object center" refer to the geometric centroid or the center of mass (where the former denotes the center of the bounding box and the latter denotes the mean coordinate of all points belonging to the object)? Given the inherent sparsity of point clouds, these two centers may exhibit significant differences.
3. The statistical analysis of the importance of spatial information for 3D-RES should be included in the main text. This will help readers understand the motivation of the paper from both qualitative and quantitative perspectives.
4. Although this paper discusses its limitations, it does not provide failure cases or corresponding analyses. Specifically, for the segmentation of plural nouns, it remains unclear whether only one object is recognized or if segmentation fails altogether. It would be beneficial to include either qualitative or quantitative analysis in this regard. Including this part would make the paper more comprehensive and facilitate follow-up research and improvements.

**Questions:**

1. Does "object center" in this paper refer to the geometric centroid (center of the bounding box) or the center of mass (mean coordinate of all points)?

**Limitations:**

The authors have discussed the limitations, and I think it is somewhat okay for this work. It would be even better if an analysis of bad cases could be included.

---

> ### Author Rebuttal · Authors · 2024-08-06
>
> Thank you for your positive feedback and acknowledgment of our paper's strengths. We're pleased you appreciate our approach of attributing 3D spatial properties to text for 3D multimodal spatial perception modeling, recognize the promising performance of our model on ScanRefer, and underscore the effectiveness of our proposed methods. Now, we'll address the specific concerns you've raised to provide further clarification:
>
> ---
>
> > **Q1: RG-SAN adopts superpoints as fundamental visual units for feature extraction and segmentation. While previous works have employed similar approaches, I am interested in understanding the segmentation quality of superpoints themselves. For instance, how many superpoints exist solely within individual objects? This is crucial because if a superpoint spans across two objects, it inevitably affects the segmentation results.**
> >
>
> A1: Thank you for your insightful feedback. In practice, due to the fine granularity of superpoints and their tendency to aggregate semantically similar points, most superpoints cover only a single object. To verify this, we conducted a statistical analysis of the superpoints in the ScanRefer [5] dataset. If a superpoint contains points from more than one object, it is classified as containing multiple objects; otherwise, it is categorized as containing a single object. Our analysis reveals that 99.55% of the points are within superpoints that cover a single object, with a missing probability of less than 0.5%. This indicates that the issue of multiple objects within a single superpoint has a negligible impact on the final results and does not warrant special attention.
>
> ---
>
> > **Q2: RG-SAN trains spatial awareness networks using the centroid coordinates of target objects. Here, does "object center" refer to the geometric centroid or the center of mass (where the former denotes the center of the bounding box and the latter denotes the mean coordinate of all points belonging to the object)? Given the inherent sparsity of point clouds, these two centers may exhibit significant differences.**
> >
>
> A2: Thank you for your valuable comments. We use the centroid of all superpoints belonging to an object, representing the average coordinates of these superpoints. We conducted comparative experiments between this centroid setting and another centroid setting. As shown in Table A, the results indicate no significant differences between the two approaches.
>
> We will include this discussion in the revised version to enhance the clarity of the paper.
>
> | Setting | Unique mIoU | Multiple mIoU | Overall mIoU |
> | --- | --- | --- | --- |
> | Center of Box | 74.8 | 37.4 | 44.7 |
> | Center of Mass | 74.5 | 37.4 | 44.6 |
>
> Table A: Comparison of the Center of Box and Center of Mass.
>
> ---
>
> > **Q3: The statistical analysis of the importance of spatial information for 3D-RES should be included in the main text. This will help readers understand the motivation of the paper from both qualitative and quantitative perspectives.**
> >
>
> A3: Thank you for your suggestion. We will include the statistical analysis on the importance of spatial information for 3D-RES in the paper to enhance the rigor of the paper.
>
> ---
>
> > **Q4: Although this paper discusses its limitations, it does not provide failure cases or corresponding analyses. Specifically, for the segmentation of plural nouns, it remains unclear whether only one object is recognized or if segmentation fails altogether. It would be beneficial to include either qualitative or quantitative analysis in this regard. Including this part would make the paper more comprehensive and facilitate follow-up research and improvements.**
> >
>
> A4: Thanks for your insightful suggestion. We will include visualizations of failure cases and provide a qualitative analysis in the new version.
>
> ---
>
> > **Q5: Does "object center" in this paper refer to the geometric centroid (center of the bounding box) or the center of mass (mean coordinate of all points)?**
> >
>
> A5:  Thank you for your detailed question. We are referring to the centroid, where the center coordinate is defined as the mean of the coordinates of all superpoints belonging to the object.

---

> > ### Author Response · Authors · 2024-08-09
> > **Sincere Request for Further Discussions**
> >
> > Dear Reviewer p7dx,
> >
> > Thanks again for your great efforts and constructive advice in reviewing this paper! With the discussion period drawing to a close, we expect your feedback and thoughts on our reply. We put a significant effort into our response, with several new experiments and discussions. We sincerely hope you can consider our reply in your assessment.
> >
> > We look forward to hearing from you, and we can further address unclear explanations and remaining concerns if any.
> >
> > Regards, Authors

---

> > ### Author Response · Authors · 2024-08-13
> >
> > Dear Reviewer p7dx,
> >
> > We are grateful for your thorough review and the constructive feedback provided on our submission. Your insights have significantly contributed to the refinement of our paper. We have endeavored to address all the points raised in your initial review comprehensively.
> >
> > As the discussion period for NeurIPS 2024 is drawing to a close, we would appreciate knowing if there are any further clarifications or additional details you might need. We are fully prepared to continue discussions to further enhance the quality of our work.
> >
> > With appreciation,
> >
> > Paper 9950 Authors

---

> > > ### Comment · Reviewer_p7dx · 2024-08-13
> > > **Comment to Author**
> > >
> > > Thank you for the detailed and thorough response. The replies have addressed all of my concerns. The in-depth discussion on superpoints and centroid coordinates will further enhance the generalizability and reproducibility of the paper's conclusions.
> > >
> > > I have also carefully reviewed the responses and opinions of other reviewers. It is evident that the reviewers generally acknowledge the contribution of modeling spatial positions within the language space. Even though Reviewer 6dEW still has some reservations about the position encoding, the author's appropriate and detailed responses have effectively addressed these concerns, which I find convincing. Therefore, I will maintain my score for this paper.

---

> > > > ### Author Response · Authors · 2024-08-14
> > > >
> > > > Thank you for your thorough review and valuable feedback. Your suggestions will greatly improve our paper. We will incorporate your feedback into the final version and make the code open source for the community to learn from. Once again, we sincerely appreciate your input.

---

### Official Review · Reviewer_6dEW · 2024-07-08

**Soundness:** 2
**Presentation:** 2
**Contribution:** 2
**Rating:** 3
**Confidence:** 4

**Summary:**

The paper proposes a new framework for 3D referring expression segmentation. The main contributions include analyzing the spatial information among objects and rule-guided target selection. Extensive experiments validate the effectiveness of the proposed method.

**Strengths:**

The authors develop the method to achieve the state of the art performance on ScanRefer benchmark for 3D Referring Expression Segmentation. The authors conduct detailed experiments and comparison to validate the design.

**Weaknesses:**

1. Both the idea of incorporating the spatial information is not new. For spatial relation, section 3.2.2 and section 3.2.3 are very similar to [22]. Equation (7) to equation (10) are almost identical to equation (5) to equation (7) in [22]. Equation (12) is similar to equation (8) in [22]. What's more, there have been a lot of work on spatial information in 3D visual grounding, such as [6].
2. The writing needs improvement. Some notations are not well explained. For example, K_i in line 120 and c_i in line 123 are not introduced. P^t in equation (3) with two subscripts is inconsistent with P^t in equation (5). Table t in line 166 and q in equation (7) are not introduced.

**Questions:**

1. The text-driven localization module is similar to the architecture in [22]. Have you tried to report their performance in ScanRefer in table 1?
2. In equation (2), how do you initialize W_E and W_S to obtain the initial representations?
3. In terms of equation (5), what is the intuition of adding positional encoding to position features? Does the order of the visual feature affects the final attention output?
4. In equation (6) the positional encoding is added while in equation (9) and (10) it is concatenated. Why are these different?
5. For section 3.3.1, do you have separate evaluation on how your algorithm performs in terms of finding the target?
6. The explanation for table 4 points out the Top1 tends to select different nodes variably (line 281), but RTS is also choosing different nodes?

**Limitations:**

The authors addresses the limitations and societal impact.

---

> ### Author Rebuttal · Authors · 2024-08-06
>
> > Q1: Using spatial information is not new. Sec. 3.2.2 and 3.2.3 and Eq. (7) to (12) resemble those in [22]. Spatial information in 3D visual grounding has been explored, like [6].
> >
>
> A1: Thank you for your valuable feedback. Indeed we acknowledge that our use of positional encoding is based on previous methods[51][22]. However, our primary contribution lies in modeling the spatial positions and relationships of noun entities within sentences for the 3D-RES task, which has also been recognized by reviewers p7dx, 8gkL, and MnE2. This approach has two core differences compared to [22]:
>
> (1) Unlike [22], which uses zero initialization for queries and random initialization for positional information, our queries and positions are text-driven from the start. This improves mIoU by 4.4 points, as shown in Tab. 2 and the newly added Tab. A.
>
> (2) Unlike [22], which supervises all target instances' positions, in 3D-RES, only the core target word is supervised. Our novel RWS method constructs spatial relationships for all noun instances using only the target word's positional information, improving mIoU by 2.3 points, as shown in Tab. 4.
>
> | Method | Initialization method of Queries | Initialization method of Position | Multiple mIoU | Overall mIoU |
> | --- | --- | --- | --- | --- |
> | MAFT [22] | Zero | Random | 29.7 | 37.9 |
> |  | Text-driven | Random | 30.1 | 38.8 |
> | RG-SAN w/o RWS | Text-driven | Text-driven | 34.7 | 42.3 |
> | RG-SAN (Ours) | Text-driven | Text-driven | 37.4 | 44.6 |
>
> Table A: Comparison of MAFT [22] with our RG-SAN in ScanRefer Dataset.
>
> **In summary, our core innovation lies in constructing spatial positional information, rather than just using positional encoding, as done in [6].** Using positional encoding is a regular operation after generating spatial information. We also explored other positional encodings, such as 5D Euclidean RPE, achieving similar results in Tab. 2.
>
> Following your suggestion, we will further compare and analyze our work with the highly relevant and interesting study [22] in the new version to clarify our contributions.
>
> ---
>
> > **Q2: The writing needs improvement: K_i (line 120) and c_i (line 123) are not defined, P^t in Eq. (3) with two subscripts is inconsistent with P^t in Eq. (5), and Table t (line 166) and q (Eq. 7) are not introduced.**
> >
>
> A2: Thank you for your detailed feedback. We adopted the representation from [28] to formulate our approach as concisely as possible. Following your suggestion, we will make improvements in new version.
>
> ---
>
> > **Q3: The TLM module resembles [22]'s architecture. Have you compared their performance on ScanRefer in Tab. 1?**
> >
>
> A3: Thank you for your constructive suggestions. We conducted a detailed comparison with [22] in Q1A1 and reported the suggested performance. Our proposed RG-SAN improves by 6.7 points, demonstrating its effectiveness. We will include this discussion in new version. Your suggestions will enhance the robustness of our contributions.
>
> ---
>
> > **Q4: In Eq. (2), how do you initialize W_E and W_S to obtain initial representations?**
> >
>
> A4: Thank you for your attention to the details of our paper. In Eq. (2), W_E and W_S are initialized randomly. We will include this information in new version to enhance clarity.
>
> ---
>
> > **Q5: For Eq. (5), what is the intuition of adding positional encoding to position features? Does the order of visual features impact the attention output?**
> >
>
> A5:  Thank you for your constructive question. Adding absolute position encoding is common in computer vision [51]. Changing the input order of visual features does not affect the final attention output because (1) the attention mechanism is order-invariant, and (2) the position encoding is tied to the visual tokens' 3D positions (xyz), so altering input order does not impact these positions.
>
> ---
>
> > **Q6: Why is positional encoding added in Eq. (6) but concatenated in Eqs. (9) and (10)?**
> >
>
> A6: Thank you for your insightful inquiry. Our experiments show that addition and concatenation for positional encoding in Eq. (6), (9), and (10) yield similar results, as shown in Tab. B. The differences are negligible, so either approach can be used without significantly impacting the outcome.
>
> | Eq. (6) | Eqs. (9), (10) | Unique mIoU | Multiple mIoU | Overall mIoU |
> | --- | --- | --- | --- | --- |
> | Cat |  Cat | 74.6 | 37.4 | 44.6 |
> | Cat | Add | 74.7 | 37.4 | 44.7 |
> | Add | Cat | 74.5 | 37.4 | 44.6 |
> | Add | Add | 75.1 | 37.5 | 44.8 |
>
> Table B: Ablation of positional encoding usage, where "Cat" denotes concatenation, while "Add" denotes direct addition.
>
> ---
>
> > **Q7: Evaluation of RTS for finding the target.**
> >
>
> A7: Thank you for your question. We previously evaluated RTS's ability to find the target using LLAMA2 70B in Sec. F of the supplementary materials, achieving an 80% match rate. However, LLAMA2 is not entirely accurate, making it an unreliable benchmark.
>
> To better validate RTS, we annotated the text of 9,508 Val set samples to mark the target word positions. RTS achieved an accuracy of 93.4%, compared to 63.7% for the Top1 method, confirming our algorithm's effectiveness.
>
> We will include this evaluation in new version and open-source the annotations to ensure reproducibility.
>
> ---
>
> > **Q8: The explanation for Tab. 4 points out Top1 tends to select different nodes variably (line 281), but RTS is also choosing different nodes?**
> >
>
> A8: We apologize for the confusion. In line 281, "different" refers to predicted nodes that are not the target noun word. Top1 often selects nodes other than the target word, such as adjectives or verbs, leading to semantic confusion and an accuracy of only 63.7%.
>
> In contrast, RTS accurately identifies the target word based on syntax, regardless of its position, achieving an accuracy of 93.4% as Q7A7 points out. This precise selection enhances semantic accuracy and significantly improves performance, as shown in Tab. 4.
>
> We will revise the description in new version to improve clarity.

---

> > ### Author Response · Authors · 2024-08-09
> > **Sincere Request for Further Discussions**
> >
> > Dear Reviewer 6dEW,
> >
> > Thanks again for your great efforts and constructive advice in reviewing this paper! With the discussion period drawing to a close, we expect your feedback and thoughts on our reply. We put a significant effort into our response, with several new experiments and discussions. We sincerely hope you can consider our reply in your assessment.
> >
> > We look forward to hearing from you, and we can further address unclear explanations and remaining concerns if any.
> >
> > Regards,
> > Authors

---

> ### Comment · Reviewer_6dEW · 2024-08-12
>
> Thanks for the detailed rebuttal. The authors have addressed some of my concerns. Here are some feedback:
>
> For Q2: You should not follow the notations from other work and assume the readers could follow. Please clearly define the notations in the revised version.
>
> For Q5: If the order of visual features should not affect the output, then you should **not** add positional encoding to the visual features. Same to the positional features. If you add positional encoding, then the attention output would change if the inputs are permuted. This is a technical flawless.
>
> For Q6: I hope you could be consistent about how you deal with the positional encoding (if you need to add it). Looks from the new ablation result using 'add' for both lead to the best performance.
>
> I have also read the reviews from other reviewers. I would maintain my original rating as some of my concerns are not well-addressed.

---

> > ### Author Response · Authors · 2024-08-13
> > **Response to Reviewer 6dEW (part-1)**
> >
> > Thank you very much for your prompt, positive, and clear feedback. We greatly appreciate your recognition of our core motivation and the primary technological innovations. We will now address the remaining concerns you raised, including those related to the presentation and the positional encoding aspect. We hope our forthcoming responses will meet your expectations.
> >
> > > Feedback1: For Q2: You should not follow the notations from other work and assume the readers could follow. Please clearly define the notations in the revised version.
> > >
> >
> > **FA1:** Thank you for your constructive feedback. We are committed to addressing your comments and will refine the revised version to improve the clarity of the notations. Your input is invaluable to us, and we appreciate your guidance in helping us strengthen our work.
> >
> > ---
> >
> > > Feedback2: For Q5: If the order of visual features should not affect the output, then you should **not** add positional encoding to the visual features. Same to the positional features. If you add positional encoding, then the attention output would change if the inputs are permuted. This is a technical flawless.
> > >
> >
> > **FA2:** Thank you very much for your feedback. To clarify this concern further, we will explain the rationale behind positional encoding to ensure better understanding.
> >
> > **(1) Input Order vs. Positional Information:**
> >
> > Firstly, changing the input order is not equivalent to altering the positional information of the inputs. Therefore, stating that the output remains unaffected by changing the input order does not imply that positional encoding is irrelevant. In fact, as shown in Table 3 of our paper, incorporating appropriate positional encoding can improve performance by 0.6 to 1.2 points. Although this improvement may not be as significant as the gains from our core module, positional encoding remains a classic operation in computer vision. We have retained this module as a byproduct of modeling positional information in our work. We will now provide examples to illustrate the difference between altering input order and positional information in the follow.
> >
> > **(2) 2D Positional Encoding vs. 3D Positional Encoding:**
> >
> > Unlike 2D positional encoding, which typically uses indices, 3D point clouds exhibit unordered and sparse characteristics, making index-based encoding unsuitable. Instead, 3D positional encoding employs Fourier encodings of the 3D coordinates (xyz), where `xyz` represents the spatial positions of the point cloud relative to the scene center (0, 0, 0). We will further illustrate the difference between these two approaches and how input order affects them in the following **part-2**.

---

> > ### Author Response · Authors · 2024-08-13
> > **Response to Reviewer 6dEW (part-2)**
> >
> > ## **Supplement to FA2:**
> >
> > ### **Example of 2D Positional Encoding:**
> >
> > Assume an input image of size 3x3:
> >
> > | Patch0 (0, 0) | Patch1  (0, 1) | Patch2 (0, 2) |
> > | --- | --- | --- |
> > | Patch3 (1, 0) | Patch4 (1, 1) | Patch5 (1, 2) |
> > | Patch6 (2, 0) | Patch7 (2, 1) | Patch8 (2, 2) |
> >
> > In a Vision Transformer (ViT), positional encoding (PosEmb) is based on token indices:
> >
> > - PosEmb(0, 0) + Patch 0 -> [Final Embedding 0]
> > - PosEmb(0, 1) + Patch 1 -> [Final Embedding 1]
> > - PosEmb(0, 2) + Patch 2 -> [Final Embedding 2]
> > - PosEmb(1, 0) + Patch 3 -> [Final Embedding 3]
> > - PosEmb(1, 1) + Patch 4 -> [Final Embedding 4]
> > - PosEmb(1, 2) + Patch 5 -> [Final Embedding 5]
> > - PosEmb(2, 0) + Patch 6 -> [Final Embedding 6]
> > - PosEmb(2, 1) + Patch 7 -> [Final Embedding 7]
> > - PosEmb(2, 2) + Patch 8 -> [Final Embedding 8]
> >
> > Assume the modified input order is as follows:
> >
> > | Patch8 (0, 0) | Patch0 (0, 1) | Patch7 (0,2) |
> > | --- | --- | --- |
> > | Patch1 (1, 0) | Patch6 (1, 1) | Patch2 (1, 2) |
> > | Patch5 (2, 0) | Patch3 (2, 1) | Patch4 (2, 2) |
> >
> > The final features will be:
> >
> > - **PosEmb(0, 0) + Patch 8** -> [Final Embedding 0]
> > - **PosEmb(0, 1) + Patch 0** -> [Final Embedding 1]
> > - **PosEmb(0, 2) + Patch 7** -> [Final Embedding 2]
> > - **PosEmb(1, 0) + Patch 1** -> [Final Embedding 3]
> > - **PosEmb(1, 1) + Patch 6** -> [Final Embedding 4]
> > - **PosEmb(1, 2) + Patch 2** -> [Final Embedding 5]
> > - **PosEmb(2, 0) + Patch 5** -> [Final Embedding 6]
> > - **PosEmb(2, 1) + Patch 3** -> [Final Embedding 7]
> > - **PosEmb(2, 2) + Patch 4** -> [Final Embedding 8]
> >
> > If the order of the input tokens is changed,  the final embeddings will differ, leading to different outcomes.
> >
> > ### **Example of 3D Positional Encoding:**
> >
> > Assume a point cloud with the following data:
> >
> > | Point Index | xyz | rgb |
> > | --- | --- | --- |
> > | Point 1 | (1.0, 2.0, 3.0) | (255, 0, 0) |
> > | Point 2 | (4.0, 5.0, 6.0) | (0, 255, 0) |
> > | Point 3 | (7.0, 8.0, 9.0) | (0, 0, 255) |
> > | Point 4 | (1.5, 2.5, 3.5) | (255, 255, 0) |
> > | Point 5 | (4.5, 5.5, 6.5) | (255, 0, 255) |
> > | Point 6 | (7.5, 8.5, 9.5) | (0, 255, 255) |
> > | Point 7 | (2.0, 3.0, 4.0) | (128, 128, 128) |
> > | Point 8 | (5.0, 6.0, 7.0) | (64, 64, 64) |
> > | Point 9 | (8.0, 9.0, 10.0) | (192, 192, 192) |
> >
> > **Where:**
> >
> > - The `xyz` column represents the coordinates (x, y, z) of the point cloud, indicating the three-dimensional spatial positions relative to the scene center (0, 0, 0) in the scene coordinate system.
> > - The `rgb` column denotes the color (r, g, b) of the point cloud.
> >
> > An example of the 3D absolute positional encoding is as follows:
> >
> > | Point Index | xyz | Point Feature | Positional Encoding | Final Embedding |
> > | --- | --- | --- | --- | --- |
> > | Point 1 | (1.0, 2.0, 3.0) | PointFeat 1 | PosEmb(1.0, 2.0, 3.0) | PointFeat 1 + PosEmb(1.0, 2.0, 3.0) |
> > | Point 2 | (4.0, 5.0, 6.0) | PointFeat 2 | PosEmb(4.0, 5.0, 6.0) | PointFeat 2 + PosEmb(4.0, 5.0, 6.0) |
> > | Point 3 | (7.0, 8.0, 9.0) | PointFeat 3 | PosEmb(7.0, 8.0, 9.0) | PointFeat 3 + PosEmb(7.0, 8.0, 9.0) |
> > | Point 4 | (1.5, 2.5, 3.5) | PointFeat 4 | PosEmb(1.5, 2.5, 3.5) | PointFeat 4 + PosEmb(1.5, 2.5, 3.5) |
> > | Point 5 | (4.5, 5.5, 6.5) | PointFeat 5 | PosEmb(4.5, 5.5, 6.5) | PointFeat 5 + PosEmb(4.5, 5.5, 6.5) |
> > | Point 6 | (7.5, 8.5, 9.5) | PointFeat 6 | PosEmb(7.5, 8.5, 9.5) | PointFeat 6 + PosEmb(7.5, 8.5, 9.5) |
> > | Point 7 | (2.0, 3.0, 4.0) | PointFeat 7 | PosEmb(2.0, 3.0, 4.0) | PointFeat 7 + PosEmb(2.0, 3.0, 4.0) |
> > | Point 8 | (5.0, 6.0, 7.0) | PointFeat 8 | PosEmb(5.0, 6.0, 7.0) | PointFeat 8 + PosEmb(5.0, 6.0, 7.0) |
> > | Point 9 | (8.0, 9.0, 10.0) | PointFeat 9 | PosEmb(8.0, 9.0, 10.0) | PointFeat 9 + PosEmb(8.0, 9.0, 10.0) |
> >
> > **Where:**
> >
> > - The **Point Cloud Feature (PointFeat)** column represents the features of the point cloud using `PointFeat`.
> > - The **Positional Encoding (PosEmb)** column displays the positional encoding for each point in the point cloud, where `PosEmb` denotes the positional encoding function.
> > - The **Final Feature (PointFeat + PosEmb)** column illustrates the combined result of the point cloud features and the positional encoding, represented as `PointFeat` plus `PosEmb`.
> >
> > In 3D encoding, even if the input order is changed, each point’s representation remains consistent, and thus, the final output remains unchanged. However, positional information is still inherently embedded within the point cloud. We will incorporate this discussion into the revised supplementary material to make the explanation clearer.

---

> > ### Author Response · Authors · 2024-08-13
> > **Response to Reviewer 6dEW (part-3)**
> >
> > > Feedback3: For Q6: I hope you could be consistent about how you deal with the positional encoding (if you need to add it). Looks from the new ablation result using 'add' for both lead to the best performance.
> > >
> >
> > **FA3:** Thank you for your feedback. Previously, we focused primarily on the Overall mIoU metric, where the differences were indeed minimal. We appreciate you pointing this out, and as a result, we will update both operations to the "Add" setting in our final version. Your suggestion will help make our paper more robust.
> >
> > In summary, your suggestions have been incredibly helpful to us. On a broader scale, your input has clarified our motivation and technical innovations. On a detailed level, your attention to specifics has made our paper more rigorous and solid. We sincerely appreciate your contributions to improving this work. If there are any other questions or areas you'd like to discuss, we welcome further conversation.

---

> > > ### Comment · Reviewer_6dEW · 2024-08-13
> > >
> > > Thanks the authors for the detailed explanation. I might misunderstood the specific use of the 3D positional encoding in this paper, but I would still reserve my opinion. My major concern is still about the writing and presentation, as pointed out in the weakness section in my original review. I hope the authors could clearly formulate the notations and equations used in the paper, not just following the same notation and equations from other work without clear definition. I hope my comments and suggestions would help the authors to improve the manuscript.

---

> > > > ### Author Response · Authors · 2024-08-14
> > > >
> > > > Overall, we sincerely appreciate your patience in communicating with us and ultimately acknowledging the rationale behind our motivation and the innovation of our techniques. Regarding the issue you raised about the formula definitions, we initially opted for a simpler and more readable approach, which, as you pointed out, compromised the rigor of our work to some extent. Following your valuable suggestion, we have revised the symbols in the affected sections of the new version to include more detailed descriptions. These issues are straightforward to address, and we are grateful for your feedback, which has helped us make the paper clearer and more robust. We kindly hope you might consider adjusting the score of our paper accordingly. Once again, thank you very much for your constructive feedback.

---

### Author Rebuttal · Authors · 2024-08-06

We would like to express our gratitude to the reviewers for their valuable feedback and positive comments on our paper. Their insightful reviews have greatly contributed to improving the clarity and overall quality of our work.

We appreciate Reviewer **6dEW** $\color{red}{(Rating:\mathbf{3},\ Confidence:\mathbf{3})}$ for acknowledging the strengths of our paper. Specifically, they mention our new framework for 3D RES and highlight the analysis of spatial information among objects. They also recognized the excellent performance of our method and the thorough validation of the designed modules.

Reviewer **p7dx** $\color{red}{(Rating:\mathbf{7},\ Confidence:\mathbf{5})}$ appreciated our novel perspective of explicitly assigning spatial positions to text for 3D-language modeling and acknowledges our extensive comparative experiments with state-of-the-art methods and detailed ablation studies. Furthermore, they highlighted the clarity of our motivation and the thoroughness of our statistical analysis. Additionally, we are grateful for their recognition of the comparison of qualitative results with previous models. Such acknowledgment reinforces the validity of our research findings.

Reviewer **8gkL** $\color{red}{(Rating:\mathbf{7},\ Confidence:\mathbf{5})}$ acknowledged our in-depth exploration of text-conditioned 3D spatial perception, addressing both 3D spatial relationships and natural language structure. They appreciate the TLM module's role in enhancing performance and high inference speed in explicit spatial reasoning for 3D scenes. Furthermore, they acknowledge the data efficiency and generalization capabilities of our RWS module. Additionally, they acknowledge that our video demo and the visualizations in Figure 3 intuitively demonstrate the capabilities of our model.

Reviewer **MnE2** $\color{red}{(Rating:\mathbf{6},\  Confidence:\mathbf{5})}$ commended the clear articulation of our motivation and the quantitative analysis presented. They agree that spatial relation reasoning is crucial for understanding 3D scenes and recognize that our proposed method effectively extracts spatial relationships from complex language descriptions, enabling text-centric spatial reasoning. Additionally, they express interest in our rule-guided weak supervision strategy, which demonstrates the ability to perform localization and segmentation using natural language object nouns without any explicit supervision. The reviewer also supports our extensive experiments, which thoroughly validate the effectiveness of the proposed modules.

We sincerely thank the reviewers for recognizing these strengths, and we appreciate their positive feedback on the clarity, novelty, and effectiveness of our proposed methods. Their comments have further motivated us to address the concerns and improve the weaknesses pointed out in their reviews. We are committed to addressing their concerns and providing a detailed response in our rebuttal.

---

### Decision · Program_Chairs · 2024-09-25

**Decision:**

Accept (oral)

**Comment:**

The authors present a novel framework for 3D referring expression segmentation. In the words of one of the reviewers, "The authors' exploration of how spatial relationships in natural language correspond with 3D scenes tackles a crucial and challenging area, especially compared to purely visual 3D segmentation. Spatial and relational reasoning is one of the major hurdles in cross-modal 3D vision today, and it’s great to see the authors making strides in this direction. I’m confident this work will inspire further progress in embodied intelligence". The authors have addressed all of the reviewers' concerns, and there is agreement that this is a high-quality and well-executed work with novel technical contributions.